# Ribonucleotide incorporation in yeast genomic DNA shows preference for cytosine and guanosine preceded by deoxyadenosine

Sathya Balachander[1,7], Alli L. Gombolay[1,7], Taehwan Yang[1,7], Penghao Xu[1,7], Gary Newnam[1], Havva Keskin[1,5], Waleed M. M. El-Sayed [1,6], Anton V. Bryksin[2], Sijia Tao[3], Nicole E. Bowen [3], Raymond F. Schinazi [3], Baek Kim [3], Kyung Duk Koh[4], Fredrik O. Vannberg[1] & Francesca Storici [1✉]

Despite the abundance of ribonucleoside monophosphates (rNMPs) in DNA, sites of rNMP incorporation remain poorly characterized. Here, by using ribose-seq and Ribose-Map techniques, we built and analyzed high-throughput sequencing libraries of rNMPs derived from mitochondrial and nuclear DNA of budding and fission yeast. We reveal both common and unique features of rNMP sites among yeast species and strains, and between wild type and different ribonuclease H-mutant genotypes. We demonstrate that the rNMPs are not randomly incorporated in DNA. We highlight signatures and patterns of rNMPs, including sites within trinucleotide-repeat tracts. Our results uncover that the deoxyribonucleotide immediately upstream of the rNMPs has a strong influence on rNMP distribution, suggesting a mechanism of rNMP accommodation by DNA polymerases as a driving force of rNMP incorporation. Consistently, we find deoxyadenosine upstream from the most abundant genomic rCMPs and rGMPs. This study establishes a framework to better understand mechanisms of rNMP incorporation in DNA.

[1] School of Biological Sciences, Georgia Institute of Technology, Atlanta, GA, USA. [2] Molecular Evolution Core, Parker H. Petit Institute of Bioengineering and Bioscience, Georgia Institute of Technology, Atlanta, GA, USA. [3] Department of Pediatrics, School of Medicine, Emory University, Atlanta, GA, USA. [4] Lung Biology Center, Department of Medicine, University of California San Francisco, San Francisco, CA, USA. [5]Present address: Omega Bio-tek, Norcross, GA, USA. [6]Present address: Marine Microbiology Department, National Institute of Oceanography and Fisheries, Red Sea, Egypt. [7]These authors contributed equally: Sathya Balachander, Alli L. Gombolay, Taehwan Yang, Penghao Xu. ✉email: storici@gatech.edu

Ribonucleoside monophosphates (rNMPs) are extensively incorporated by DNA polymerases into double-stranded DNA[1–9], alter the structural and mechanical properties of DNA[10], and increase DNA fragility and mutability[11–14]. Replicative DNA-dependent DNA polymerases have strong sugar discrimination, displaying preference for deoxyribonucleotides (dNTPs) over ribonucleotides (rNTPs)[15,16]. Nevertheless, rNMPs in DNA outnumber common depurination and base oxidation sites all together by more than an order of magnitude[11]. Ribonuclease (RNase) H2 cleaves 5′ to rNMPs in DNA-initiating ribonucleotide-excision repair (RER), which is the major cellular mechanism to remove rNMPs in DNA[17]. RNase H1 also cleaves at rNMPs in DNA, but targets stretches of ≥4 rNMPs[18]. The first studies reporting rNMP identity and distribution at the genome level used two yeast backgrounds for budding yeast *Saccharomyces cerevisiae* and one for fission yeast *Schizosaccharomyces pombe*, in which the gene coding for the catalytic subunit of RNase H2 (*RNH201*) was inactivated. These reports revealed some initial features of rNMPs in DNA: widespread distribution along chromosomal and mitochondrial DNA (mtDNA), abundance of rC and rG, low level of rU in budding yeast, and biased presence of rNMPs on the leading vs. lagging strand of DNA replication in association with specific low-fidelity mutants of DNA polymerases[19–22]. Further work showed that variation in the dNTP pool affects rNMP frequency in yeast mtDNA and nuclear DNA (nDNA) of *rnh201*-null cells[23], and that RER is not active in *S. cerevisiae* mtDNA[23]. In addition, depletion of dNTP pools promoted rNMP incorporation in human mtDNA[24]. Interestingly, recent studies linked rNMP incorporation to pathology in mouse models[25–27]. Therefore, it is valuable to learn more about the preferred sites of rNMP incorporation in DNA from yeast to mammalian cells.

Only limited samples have been analyzed for the presence of genomic rNMPs; thus, it remains unknown whether there are preferred sites of rNMP incorporation in DNA and whether there is conservation and/or variation in rNMP patterns across species, and different strains of the same species. Moreover, there is scant information about rNMP distribution in wild-type RNase H2

cells. Here, we perform genome-wide mapping of rNMPs in mtDNA and nDNA of three yeast species using common laboratory strains of wild type and RNase H-mutant genotypes. We use *S. cerevisiae*, the closely related, *Saccharomyces paradoxus*, and the more distantly related species, *S. pombe*[28]. To directly capture rNMPs in DNA, we use the ribose-seq protocol[21] enhanced with increased efficiency and accuracy[29]. After high-throughput sequencing, we obtain the profile of genomic rNMP incorporation using the bioinformatics toolkit Ribose-Map[30]. By analyzing 34 ribose-seq libraries, and also comparatively examining five emRiboSeq libraries[22], we uncover conserved, new features, patterns, motifs, and hotspots of rNMPs in yeast DNA, all pointing towards a major role played by the upstream deoxyribonucleotide in rNMP incorporation.

## Results

**Biased rC and rG pattern in mtDNA of wild-type *S. cerevisiae*.** We built and analyzed 11 ribose-seq libraries with mtDNA of wild-type RNase H2 cells from six commonly utilized haploid yeast laboratory strains (Methods) using three sets of restriction enzymes (RE1, RE2, and RE3; Methods, Supplementary Tables 1 and 2, Supplementary Data 1). The percentages of rA, rC, rG, and rU among these libraries were similar, regardless of the RE set used; however, we revealed some variation among the strains (Figs. 1a and 2a). rA, followed by rC and then rG, is the most abundant rNMP found in almost all libraries examined. Interestingly, the two ribose-seq libraries prepared from strain E134 had dominant rC and lower incorporation of rG, close to the dG count of 8.55%, compared to the libraries derived from all other strains. Consistently, rU was rarely incorporated in all libraries (Fig. 2a, b, Supplementary Data 1 and 2A). Normalization of single rNMP frequencies to the nucleotide content of *S. cerevisiae* mtDNA revealed bias for rC and/or rG in all libraries over rA, and especially over rU (Fig. 2a, b, $P = 4.08E{-}05$ in A–C comparison, $P = 1.52E{-}04$ in A–G comparison, and $P = 4.08E{-}05$ both in C–U and G–U comparisons, while $P = 2.77E{-}01$ in C–G comparison, Supplementary Data 3A). These data in part reflect

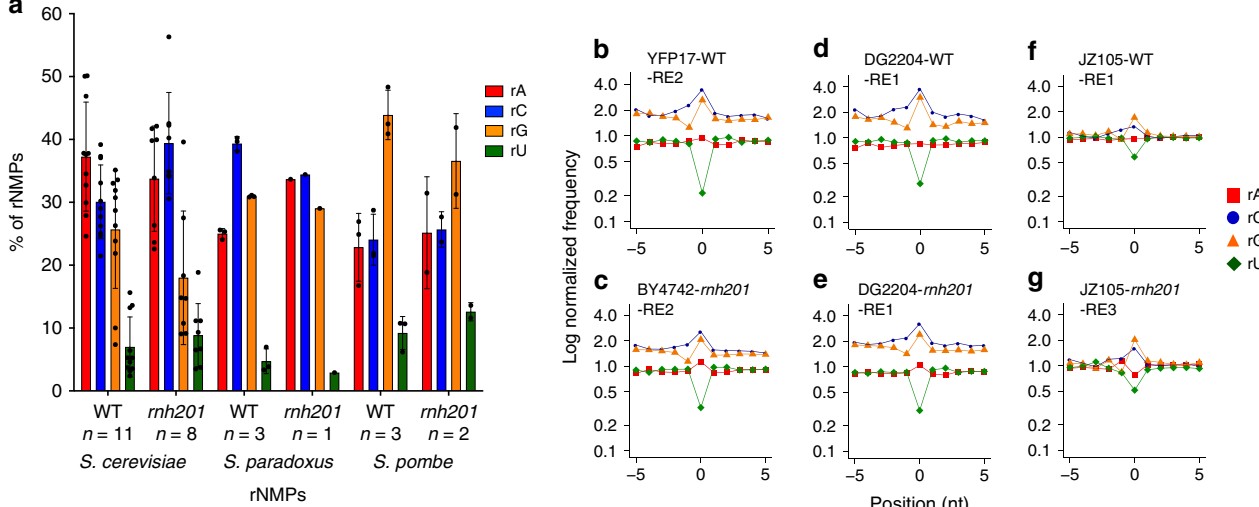

**Fig. 1 Identity and frequency of rNMP types in the mitochondrial yeast genome. a** Bar graph with corresponding data points showing percentage of rA, rC, rG, and rU found in mtDNA of WT or *rnh201*Δ strains of *S. cerevisiae*, *S. paradoxus*, and *S. pombe*. Mean, standard deviation, and number of libraries (*n*) analyzed for each genotype are shown. **b–g** Zoomed-in plots of normalized nucleotide frequencies relative to mapped positions of sequences from mitochondrial ribose-seq libraries. Plots derived from example libraries are shown for *S. cerevisiae* **b** WT (YFP17-WT-RE2-FS146) and **c** *rnh201*Δ (BY4742-*rnh201*-RE2-FS138), *S. paradoxus* **d** WT (DG2204-WT-RE1-FS108) and **e** *rnh201*Δ (DG2204-*rnh201*-RE1-FS130), and *S. pombe* **f** WT (JZ105-WT-RE1-FS94) and **g** *rnh201*Δ (JZ105-*rnh201*-RE3-FS135). Position 0 is the rNMP, − and + positions are upstream and downstream dNMPs, respectively, normalized to the A, C, G, and T content in the corresponding yeast species. Red square, A; blue circle, C; orange triangle, G; and green rhombus, U.

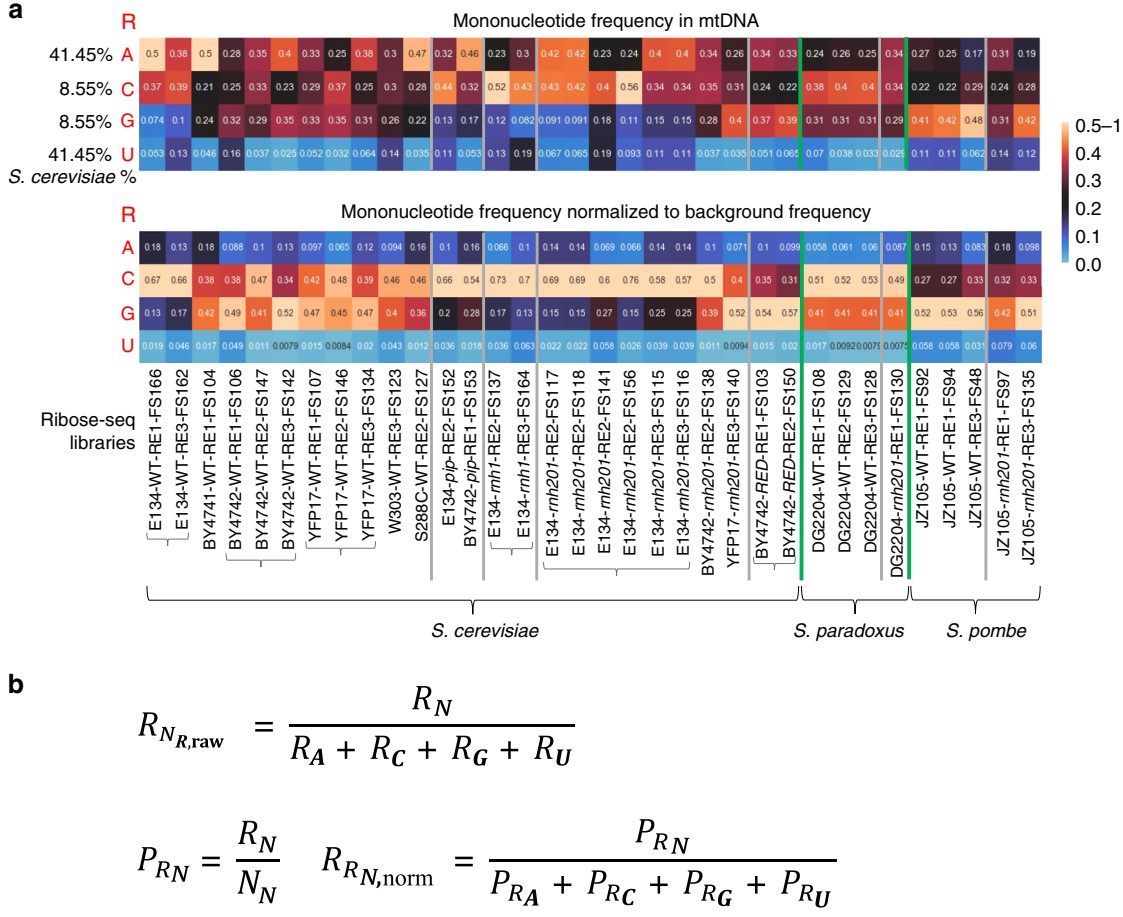

**Fig. 2 Heatmap analyses of rNMPs in yeast mitochondrial DNA. a** Heatmap analyses with (top) frequency of each type of rNMP (rA, rC, rG, and rU), and (bottom) frequency of each type of rNMP normalized to the nucleotide frequencies of the corresponding reference genome for all the mitochondrial ribose-seq libraries of this study. The corresponding formulas used are shown in **b** and explained in Methods. Each column of the heatmap shows results of a specific ribose-seq library. Each library name is indicated underneath each column of the heatmap with its corresponding strain name, genotype, and restriction enzyme (RE) set used. The yeast species of the ribose-seq libraries are also indicated. *S. cerevisiae* libraries derived from the same strains are grouped together by curly brackets. Thick, vertical, green lines separate data from the different yeast species. Vertical gray lines separate data obtained from different RNase H genotypes within each species. Each row shows results obtained for an rNMP (R in red) of base A, C, G, or U for each library. The actual percentages of A, C, G, and T bases present in mtDNA of *S. cerevisiae* are shown to the left of the top heatmap and are indicated in Supplementary Data 2A. The corresponding base percentages for mtDNA of *S. paradoxus* and *S. pombe* are indicated in Supplementary Data 2B and 2C, respectively. The bar to the right shows how different frequency values are represented as different colors: black for 0.25; black to yellow for 0.25 to 0.5–1, and black to light blue for 0.25 to 0. **b** Formulas used to calculate the frequency and the normalized frequency values of the mitochondrial heatmaps in **a**, see explanations in Methods.

the nucleotide pool imbalance, as proposed before[21] and corroborated recently[23]. By measuring rNTPs/dNTPs ratios for strains E134 and BY4742, we found that the rGTP/dGTP ratio was significantly lower in E134 compared to BY4742 ($P = 0.0015$, Supplementary Table 3, Supplementary Fig. 1), possibly accounting for the lower rG incorporation in E134. Furthermore, while proportions of rC and rG were similar in most libraries, rC was incorporated at a larger number of different sites than rG (Fig. 1b, Supplementary Fig. 2A).

We normalized the frequency by which dNMPs upstream or downstream of the rNMPs are found next to each rNMP to the A/T (41.45%) or C/G (8.55%) content. We found that in all 11 libraries rC and in part rG are located within C/G-rich areas of mtDNA, with rC being in a dC-rich, and rG in a dG-rich area (Fig. 3a, b, Supplementary Fig. A, B). This is a feature of the *S. cerevisiae* mitochondrial genome, in which dCMPs and dGMPs cluster together[31] (sacCer2 genome database). Thus, despite the low percentage of dCMPs and dGMPs in mtDNA, rCs and rGs were often surrounded by dCMPs or dGMPs, respectively.

We then studied whether rA, rC, rG, and rU were randomly incorporated in mtDNA. We reasoned that, if for example rA was randomly incorporated, the frequency by which the dNMP A, C, G, or T was found at position −1 and +1 relative to rA should reflect the frequency of the dinucleotides AA, CA, GA, TA, AC, AG, and AT obtained from the sequence of *S. cerevisiae* mtDNA (standard frequency). The frequency of CrA (Fig. 4a, Supplementary Data 2A) was above the standard value for this pair and significantly above the frequency of the other nucleotide pairs ($P = 5.35E−05$, $6.99E−05$, and $1.52E−04$ for CrA vs. ArA, GrA, and TrA, respectively, Supplementary Data 3A). For rC, rG and rU, ArC, ArG, and CrU were the highest, respectively (Fig. 4a, Supplementary Data 3A). Much less prominent difference was found among pair combinations for the dNMPs at position +1 (Supplementary Fig. 3A). We also examined the dNMPs at positions −2, +2, −3, +3, −4, and +4. Less pronouncedly than for position −1, for −2 position, we found predominant occurrences of C-rA and T-rG; for +2, rG-T; for −3, T-rG; and for −4, G-rA (Supplementary Fig. 3B–G, and Supplementary

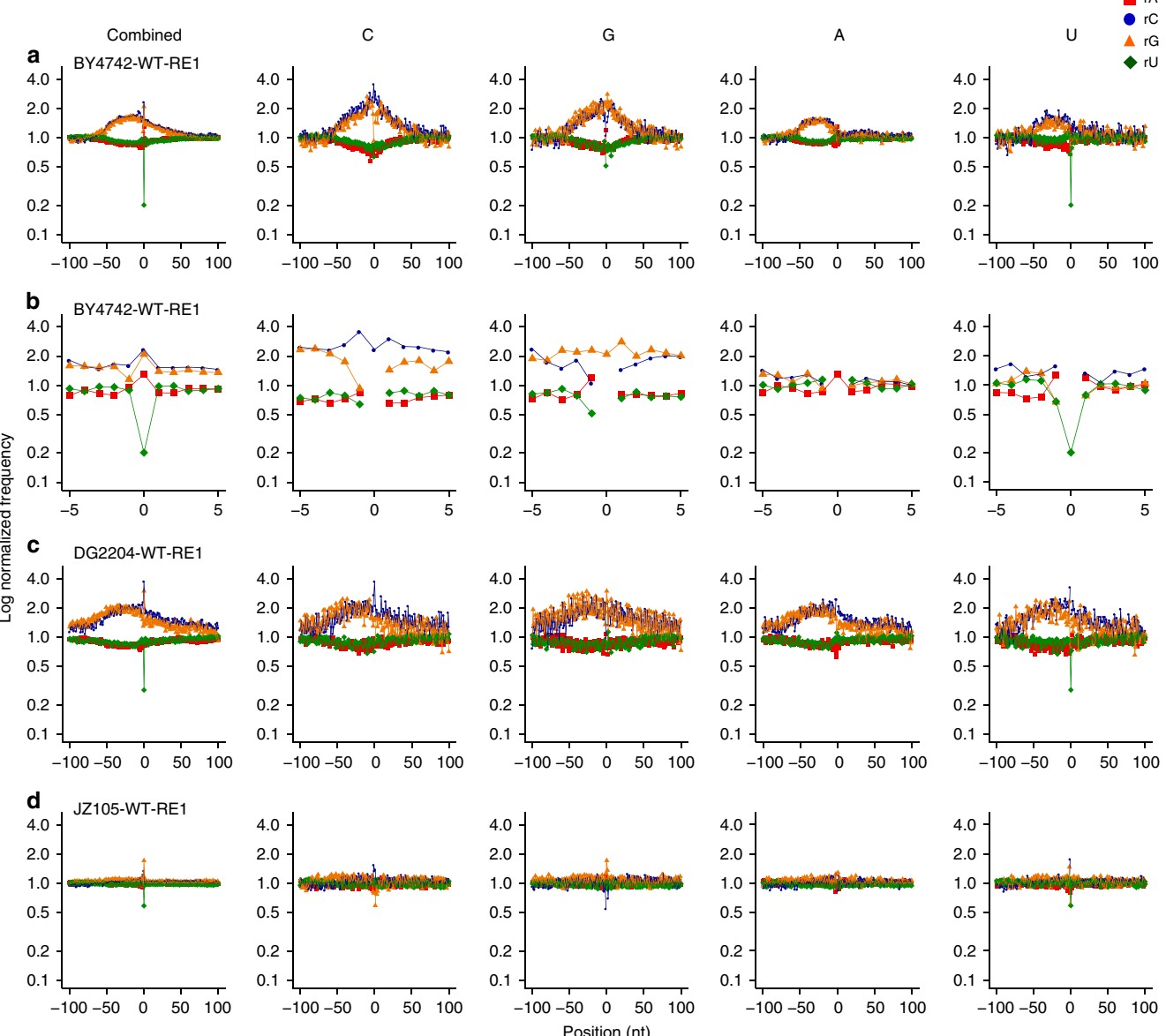

**Fig. 3 Sequence context of rNMPs in wild-type yeast mtDNA.** Shown are examples of combined and single rNMP plots of normalized nucleotide frequencies relative to mapped positions of rNMPs from examples of mitochondrial ribose-seq libraries. **a** Zoom-out and **b** zoom-in plots for mitochondrial library BY4742-WT-RE1-FS104 of *S. cerevisiae* WT cells. **c** Zoom-out plots for mitochondrial library DG2204-WT-RE1-FS108 of *S. paradoxus* WT cells. **d** Zoom-out plots for mitochondrial library JZ105-WT-RE1-FS94 of *S. pombe* WT cells. Position 0 on the *x*-axis represents the site of rNMP incorporation, − and + positions represent upstream and downstream dNMPs, respectively. The *y*-axis shows the frequency of each type of nucleotide present in the ribose-seq data normalized to the frequency of the corresponding nucleotide present in the reference genome of the indicted yeast species. Red square, A; blue circle, C; orange triangle, G; and green rhombus, U.

Data 3A). As a control, for dNMPs at position −100 or +100, the observed frequencies of dinucleotides with an rNMP matched well with those obtained from the sacCer2 mitochondrial genome (Supplementary Fig. 3H, I). Overall, these results, being also conserved among all of the 11 mitochondrial wild-type libraries, demonstrate that the dNMPs upstream from the rNMP, particularly the ones at position −1, have the most impact on the incorporation of a specific rNMP type in a given genomic position of yeast mtDNA. Moreover, these findings prove that each of the four rNMPs is not randomly incorporated in mtDNA.

**rNMP patterns in mtDNA of *S. cerevisiae* RNase H-mutant cells.** Recent records revealed lack of activity for RNase H2 in

yeast mtDNA[23,32,33]. Here, we constructed a null mutation of the catalytic subunit of RNase H2, and the ribonucleotide-excision defective (*RED*) mutations in the same subunit, impairing RER, but not RNase H2 cleavage at long RNA-DNA hybrids[34]. We built eight ribose-seq libraries of *S. cerevisiae* mtDNA derived from *rnh201*-null cells of strains E134, BY4742, and YFP17, and two libraries with the *RED* mutations derived from BY4742. As expected, the mitochondrial *rnh201*-null libraries had rNMP content similar to the mitochondrial wild-type libraries for the same strains (Figs. 1a–c and 2; Supplementary Fig. 2C, D, Supplementary Data 1). The two mitochondrial libraries derived from the *RED* cells were also similar to wild type and *rnh201* libraries except for slightly increased level of rG (Fig. 2, Supplementary Data 1). We further deleted the *RNH1* gene in E134 and

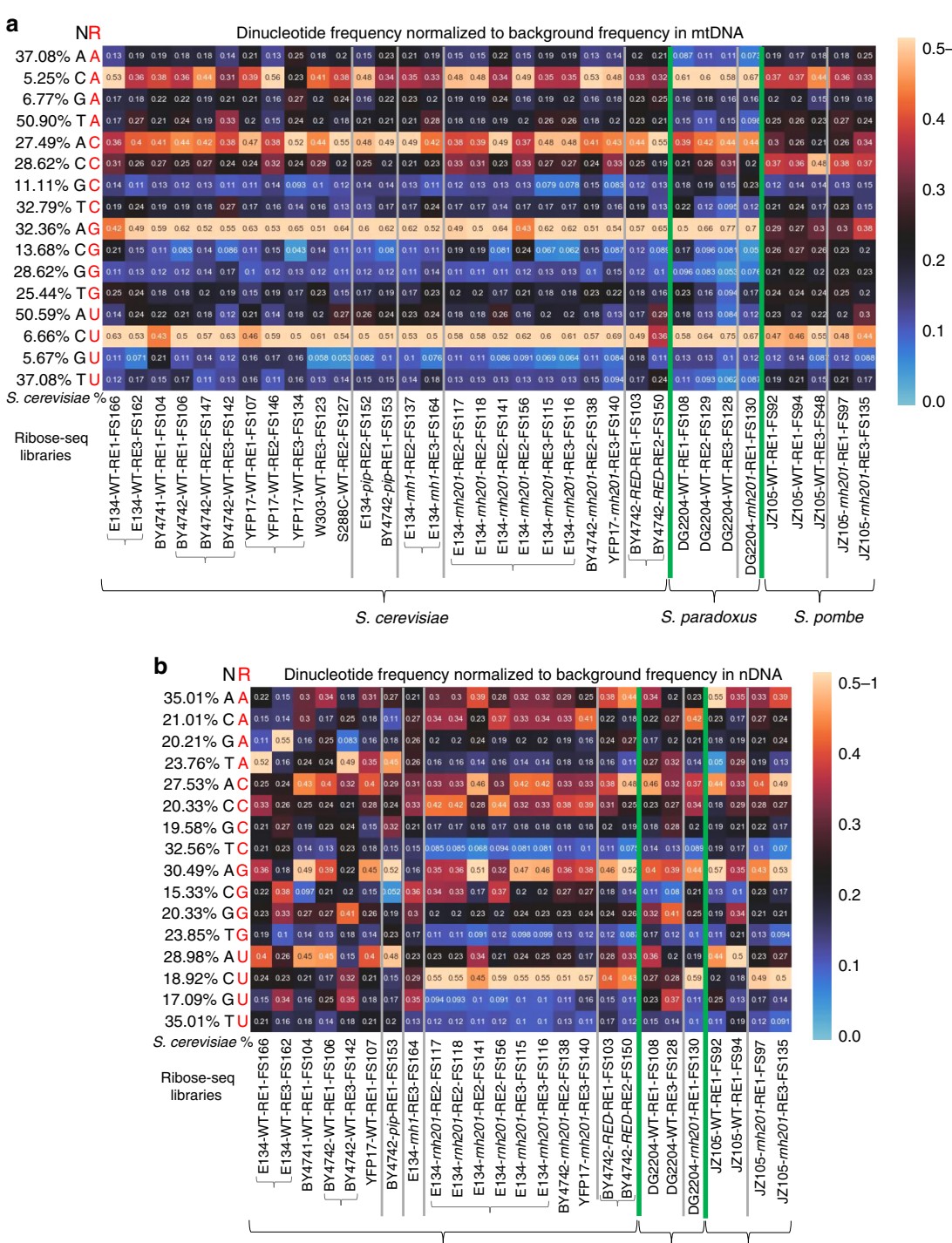

**Fig. 4 The dNMP directly upstream from each rNMP affects the frequency of rNMP incorporation.** Heatmap analyses with normalized frequency of **a** mitochondrial and **b** nuclear NR dinucleotides (rA, rC, rG, and rU with the upstream deoxyribonucleotide with base A, C, G, or T) for all the ribose-seq libraries of this study. The formulas used to calculate these normalized frequencies are shown and explained in Methods. Each column of the heatmap shows results of a specific ribose-seq library. Each library name is indicated underneath each column of the heatmap with its corresponding strain name, genotype, and restriction enzyme (RE) set used. The yeast species of the ribose-seq libraries are also indicated. *S. cerevisiae* libraries derived from the same strains are grouped together by curly brackets. Thick, vertical, green lines separate data from the different yeast species. Vertical gray lines separate data obtained from different RNase H genotypes within each species. Each row shows results obtained for a dinucleotide NR (R in red) of fixed rNMP base A, C, G, or U for each library. The actual percentages of dinucleotides of fixed base A, C, G, or T for the indicated base combinations (AA, CA, GA, and TA; AC, CC, GC, and TC; AG, CG, GG, and TG; and AT, CT, GT, and TT) present in mitochondrial and nDNA of *S. cerevisiae* are shown to the left of the heatmaps, and are also indicated in Supplementary Data 2A. The corresponding dinucleotide percentages for mitochondrial and nDNA of *S. paradoxus* and *S. pombe* are indicated in Supplementary Data 2B and 2C, respectively. The observed % of dinucleotides with rNMPs with base A, C, G, or U were divided by the actual % of each dinucleotide with fixed base A, C, G, or T in mitochondrial or nDNA of the corresponding species. The bar to the right shows how different frequency values are represented as different colors: back for 0.25; black to yellow for 0.25 to 0.5–1, and black to light blue for 0.25 to 0.

constructed two independent ribose-seq libraries. Data analysis from the *rnh1*-null libraries showed rNMP content similar to that of mitochondrial wild type and *rnh201*-null cells of E134 (Fig. 2, Supplementary Fig. 2C, D, Supplementary Data 1). Consistently with our results in wild-type RNase H2 cells, we found that CrA, ArC, ArG, and CrU were significantly above the standard count for the respective dinucleotides in mtDNA and significantly above the other dinucleotide pair combinations in all *rnh201*-null, *RED*, and *rnh1*-null cells (Fig. 4a, Supplementary Fig. 3A–I, Supplementary Data 3A). Moreover, to ensure that the results of the dinucleotides were not artifacts of the ribose-seq technique, using Ribose-Map[30], we analyzed data from five libraries of *S. cerevisiae* *rnh201*-null cells prepared with the emRiboSeq technique, which does not employ restriction enzymes, alkali, and AtRNL, but sonication, human recombinant RNase H2, and T4-quick ligase, respectively[22]. We generated heatmaps for the mononucleotides and examined the dNMPs at positions −1, +1, −2, and +2 relative to the rNMPs in these libraries (Supplementary Fig. 4A, C, E, G, H). The similarities between these emRiboSeq and the ribose-seq results are remarkable. This comparison further supports our findings that the dNMP upstream of the rNMP has the most impact on the incorporation of a specific rNMP type in yeast mtDNA.

**Patterns of rNMPs in *S. paradoxus* and *S. pombe* mtDNA.** To determine whether the patterns of rNMP incorporation in mtDNA vary in different yeast species, we built ribose-seq libraries from mtDNA of wild type and *rnh201*-null *S. paradoxus*, a species that is evolutionarily close to *S. cerevisiae*, and *S. pombe*, which is evolutionarily more distant from *S. cerevisiae*[28]. We analyzed three ribose-seq libraries with mtDNA of wild type and one from *rnh201*-null *S. paradoxus* of strain DG2204, as well as three ribose-seq libraries with mtDNA of wild type and two from *rnh201*-null *S. pombe* of strain JZ105 (Supplementary Table 1, Supplementary Data 1). The rNMP patterns in mtDNA of *S. paradoxus* were comparable to that of *S. cerevisiae*, being quite similar to strains YFP17, W303, and S288C, displaying a preference for rC and low incorporation of rU (Figs. 1a, d, e and 2a, b; Supplementary Data 1, 2B and 3A). *S. pombe* mtDNA had higher presence of rG, both in wild type and *rnh201*-null cells, while still displaying low rU (Figs. 1a, f, g and 2a, b, Supplementary Data 1, 2C and 3A). Like in *S. cerevisiae* cells, the rUTP/dTTP ratio was the lowest among rNTPs/dNTPs ratios in *S. pombe* strain JZ105 (Supplementary Fig. 1). We did not detect major differences in the rNMP frequencies between wild type and *rnh201*-null mitochondrial libraries in either *S. paradoxus* or *S. pombe* (Fig. 2a).

While rNMPs were located within C/G-rich areas of *S. paradoxus* mtDNA, similarly to *S. cerevisiae*, this was not the case for rNMPs in *S. pombe* mtDNA (Fig. 3c, d, Supplementary Fig. 2E–H), highlighting a unique feature of mitochondrial rNMPs of budding yeasts. Markedly, like in *S. cerevisiae*, we found that the dNMP upstream from the rNMP had the most impact on rNMP incorporation both in *S. paradoxus* and *S. pombe*. For *S. paradoxus*, we found that, as in *S. cerevisiae*, CrA, ArC, ArG, and CrU were the most abundant pairs. *S. pombe* mtDNA showed preference for CrA and CrU, and no preference for any dNMP following the rNMP, like *S. cerevisiae*. Differently from budding yeasts, in *S. pombe* wild-type libraries, rC showed preference for CrC (Fig. 4a, Supplementary Fig. 3A–I, Supplementary Data 2C and 3A).

**Wild-type *S. cerevisiae* nDNA has low rG and high rC.** We built six ribose-seq libraries of nDNA from wild-type RNase H2 cells of strains E134, BY4741, BY4742, and YFP17, eight libraries from

nDNA of *rnh201*-null cells from strains E134, BY4742, and YFP17, two libraries from *RED* cells of strain BY4742, one from *rnh1*-null cells of strain E134, and one from *rnh202-pip* cells of strain BY4742 (Supplementary Data 4). The rNMP content and distribution in nDNA were different from those in mtDNA of the same yeast strains. In wild-type cells, rG was consistently the least abundant rNMP in all libraries. Normalization of single rNMP frequencies to the nucleotide-base content of *S. cerevisiae* nDNA revealed bias for rC (Figs. 5a and 6a, b). While rA and rU were incorporated proportionally to the abundance of dA and dT in *S. cerevisiae* nDNA, in all of these eight libraries, rC occurred above the 19.00% C count, and its normalized content was significantly greater than that of the other rNMPs (Figs. 5b and 6; Supplementary Data 2A and 3B, and Supplementary Fig. 5A, B).

Due to the activity of RNase H2 on nDNA, the nuclear rNMP content in *rnh201*-null cells was distinct from that in wild-type cells. In nDNA of *rnh201*-null cells, rC was by far the most abundant rNMP. On average, over 50% of rNMPs were rCs. The rA fraction dropped substantially, and rU was the least abundant (Figs. 5a, c and 6, Supplementary Fig. 5C, D, Supplementary Data 4). The data were consistent among the different strains. The two *RED* libraries had increased level of rG (Fig. 6, Supplementary Fig. 5C, D, Supplementary Data 4).

In vitro studies have shown that the interaction with proliferating cell nuclear antigen (PCNA), via the PCNA-interacting peptide domain (PIP-box), enhances cleavage of misincorporated rNMPs by the archaeal RNase HII but not the yeast or human RNase H2 (refs. [34,35]). In line with these results, we found that in *rnh202-pip* mutant cells of *S. cerevisiae*, the nuclear rNMP content remained similar to that obtained in wild-type cells (Fig. 6, Supplementary Fig. 5A, B, Supplementary Data 4). Moreover, as observed in mitochondria, in *rnh1*-null cells, the rNMP content in nDNA was also similar to that in wild-type cells (Fig. 6, Supplementary Fig. 5A, B, Supplementary Data 4). These results suggest that Rnh1 does not have a strong impact on rNMP removal from nDNA.

We then examined whether rA, rC, rG, and rU were randomly incorporated in nDNA of the wild type and mutant RNase H strains by determining the frequency of the dNMPs preceding or following each rNMP, and comparing this frequency with the given frequency of each dinucleotide in *S. cerevisiae* nDNA. For rNMPs in nDNA of wild type, *rnh202-pip*, and *rnh1*-null cells, the dinucleotide pattern was less evident (Fig. 4b) than that observed in mtDNA (Fig. 4a), likely due to the lower number of rNMPs detected. Nevertheless, we found that ArC, ArG, and ArU pairs were dominant, with ArC being significantly different from GrC and TrC (Fig. 4b, Supplementary Data 2A and 3B). This effect is more evident when results obtained for the −1 are compared with those for the +1 position, and with those for the −2, +2, −3, +3, −4, +4, −100, and +100 positions, in which no particular pair was dominant across these different libraries (Supplementary Fig. 6A–I, Supplementary Data 2A and 3B). Nuclear *rnh201*-null and *RED* ribose-seq libraries showed frequencies of ArG and CrU greater than the other dinucleotide pairs for rG and rU, respectively (Fig. 4b, Supplementary Data 3B). However, differently from mtDNA, we found low frequency of dT upstream of any rNMP (Fig. 4b). Moreover, while in mtDNA CrA and ArC also stood out among the dNMPs upstream of rA and rC, respectively, these two pairs were abundant but not dominant in *rnh201*-null and *RED* libraries, showing frequencies similar to ArA and CrC, respectively (Fig. 4b, Supplementary Data 3B). These results show that rNMP incorporation in nDNA of mutant RNase H cells is not random. We did not detect major preference for the +1, −2, +2, −3, +3, −4, and +4 dNMP in *rnh201*-null and *RED* libraries (Supplementary Fig. 6A–I, Supplementary Data 3B). Analysis of the five

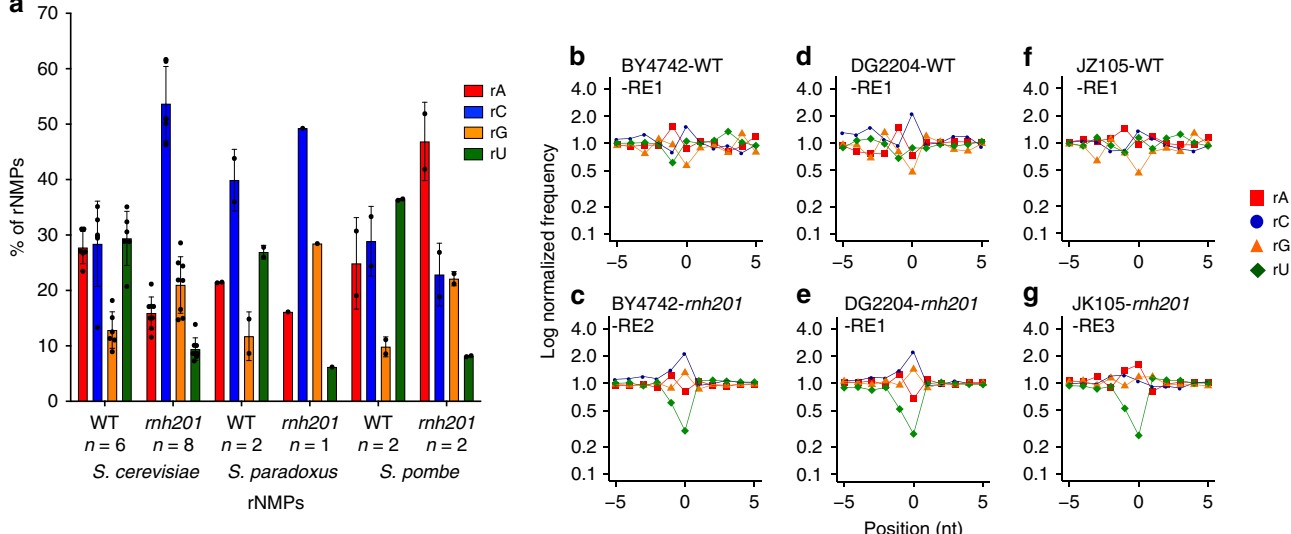

**Fig. 5 Identity and frequency of rNMP types in the nuclear yeast genome. a** Bar graph with corresponding data points showing percentage of rA, rC, rG, and rU found in nDNA of WT or *rnh201Δ* strains of *S. cerevisiae*, *S. paradoxus*, and *S. pombe*. Mean, standard deviation, and number of libraries analyzed for each genotype are shown. **b–g** Zoomed-in plots of normalized nucleotide frequencies relative to mapped positions of sequences from nuclear ribose-seq libraries. Plots derived from example libraries are shown for *S. cerevisiae* **b** WT (BY4742-WT-RE1-FS106) and **c** *rnh201Δ* (BY4742-*rnh201*-RE2-FS138), *S. paradoxus* **d** WT (DG2204-WT-RE1-FS108) and **e** *rnh201Δ* (DG2204-*rnh201*-RE1-FS130), and *S. pombe* **f** WT (JZ105-WT-RE1-FS94) and **g** *rnh201Δ* (JZ105-*rnh201*-RE3-FS135). Position 0 on the *x*-axis represents the site of rNMP incorporation, − and + positions represent upstream and downstream dNMPs, respectively. The *y*-axis shows the frequency of each type of nucleotide present in the ribose-seq data normalized to the frequency of the corresponding nucleotide present in the reference genome of the indicted yeast species. Red square, A; blue circle, C; orange triangle, G; and green rhombus, U.

*rnh201*-null emRiboSeq libraries, which are all from the same strain background (Δl(−2)l-7BYUNI300), revealed common trends in the sequence context of rNMPs observed in the ribose-seq libraries of the same genotype. In particular, the strongest biases were seen for the −1 dNMP (Supplementary Fig. 4B, D, F, I, J). Therefore, more strictly than what we found in mitochondrial libraries, the dNMP immediately upstream from the rNMP rather than the one downstream or further upstream or downstream has the most impact on the incorporation of a specific rNMP type in the nDNA of both wild type and RNase H-mutant *S. cerevisiae* cells.

**Patterns of rNMPs in *S. paradoxus* and *S. pombe* nDNA.** We analyzed two ribose-seq libraries from nDNA of wild type and one of *rnh201*-null *S. paradoxus* cells of strain DG2204, as well as two ribose-seq libraries from nDNA of wild type and two of *rnh201*-null *S. pombe* cells of strain JZ105 (Supplementary Data 4). The patterns of rNMP incorporation in nDNA of wild type and *rnh201*-null *S. paradoxus* were similar to those of the corresponding genotypes of *S. cerevisiae*, with rG the lowest and rC the dominant rNMP, similar frequency of rU and rA in wild type, and high rC, low rA, and very low rU in *rnh201*-null cells. For wild-type *S. pombe* cells, like for *S. cerevisiae* and *S. paradoxus*, rC was the highest and rG the lowest. In *rnh201*-null cells of *S. pombe*, while rU was still the lowest, rA was the most frequent rNMP (Figs. 5a, d–g and 6, Supplementary Fig. 5E–H, Supplementary Data 4). Interestingly, the rATP/dATP ratio was high in *S. pombe* compared to *S. cerevisiae* cells (Supplementary Table 3, Supplementary Fig. 1), possibly supporting elevated incorporation of rA in the absence of RNase H2 function in nDNA.

Data analysis of nDNA libraries of *S. paradoxus*, generally in line with results from *S. cerevisiae*, showed ArC and ArG as higher pairs in wild type, and mainly CrA, ArC, CrC, ArG, and CrU above the standard frequency in *rnh201*-null cells (Fig. 4b).

For *S. pombe*, together with ArC, ArG, and ArU, also ArA was above the standard value in wild-type cells, while in *rnh201*-null cells, ArA, ArC, ArG, and CrU were the highest (Fig. 4b). These findings show that rNMPs are not randomly incorporated in *S. paradoxus* and *S. pombe* nDNA and that factors beyond variation in nucleotide pools affect distribution and patterns of rNMP incorporation in nDNA. Although there is some variability for wild-type libraries, likely due to the lower number of detected rNMPs, the frequencies of dNMPs at positions +1, −2, +2, −3, +3, −4, and +4 for nDNA of wild type and *rnh201*-null cells of *S. paradoxus* and *S. pombe* did not deviate much from the dinucleotide frequencies found in the nDNA of these yeast species (Supplementary Fig. 6A–I). Overall, the results reveal substantial similarity in the dinucleotide patterns for nDNA among yeast *S. cerevisiae*, *S. paradoxus*, and *S. pombe*.

**Hotspots of rNMPs occur at ArC or ArG sites in yeast DNA.** The ribose-seq data analysis revealed rNMP sites that were in common among libraries and preferred rNMP sites in each library. We generated a list of rNMP sites that were shared among all the different mitochondrial or nuclear libraries of wild type or *rnh201*-null *S. cerevisiae* cells. Interestingly, the most abundant and shared rNMP sites in the mitochondrial libraries were found on the Crick (−) strand, which in most cases corresponds to the template strand for transcription. Moreover, the majority of these common sites were ArC or ArG in mitochondrial wild type (8/11 shared sites) and *rnh201*-null (15 of the top 25) libraries (Supplementary Data 5A, B). No shared sites were found among nuclear libraries of wild-type *S. cerevisiae*. Among the shared sites in the nDNA of *rnh201*-null cells, rC was dominant and, in the majority of cases, preceded by dA (17 of the top 25) (Supplementary Data 5C).

To determine whether there were overlapping features and specific signatures among rNMPs that were most frequently incorporated in mtDNA and nDNA, we selected the top 1% most

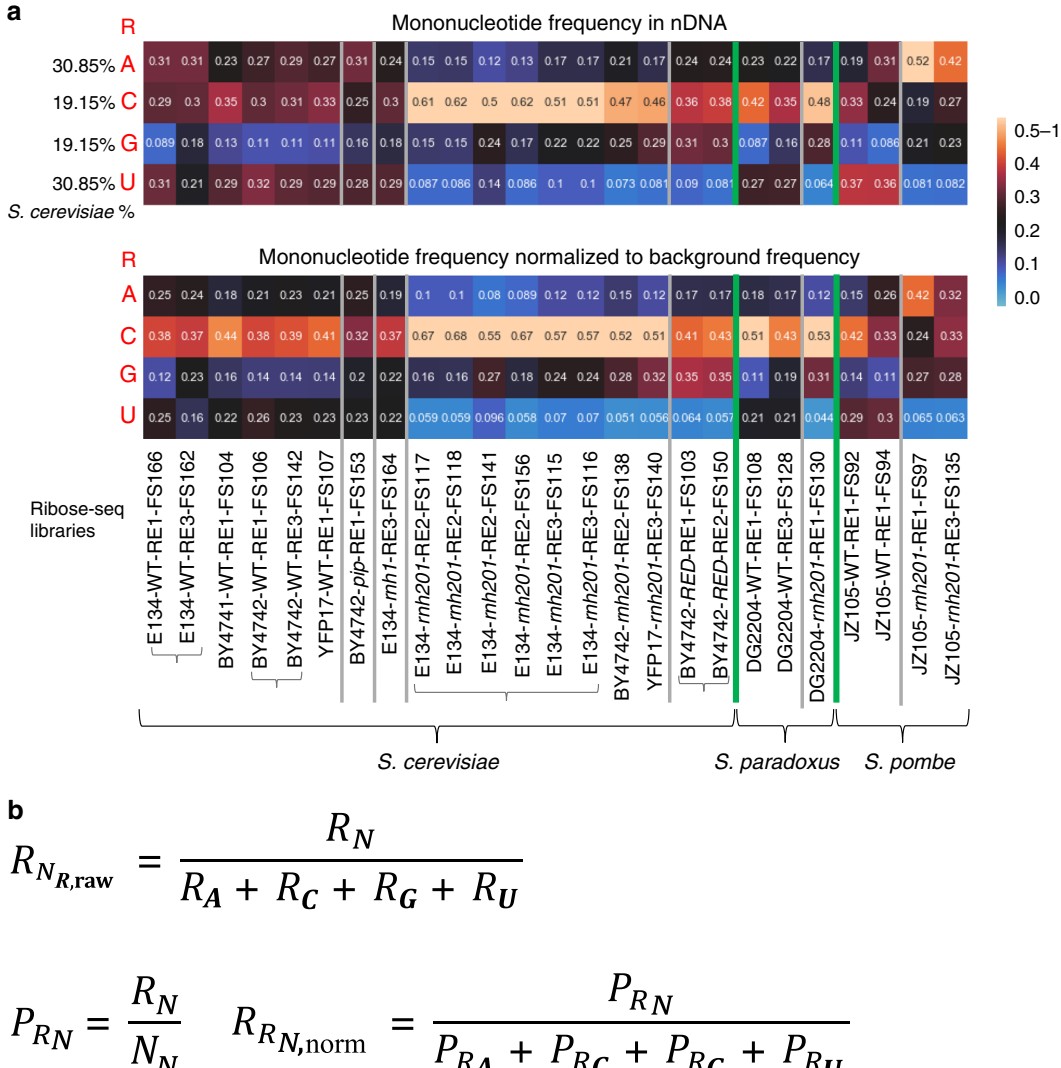

**Fig. 6 Heatmap analyses of rNMPs in yeast nuclear DNA. a** Heatmap analyses with (top) frequency of each type of rNMP (rA, rC, rG, and rU), and (bottom) frequency of each type of rNMP normalized to the nucleotide frequencies of the corresponding reference genome for all the nucleus ribose-seq libraries of this study. The corresponding formulas used are shown in **b** and explained in Methods. Each column of the heatmap shows results of a specific ribose-seq library. Each library name is indicated underneath each column of the heatmap with its corresponding strain name, genotype, and restriction enzyme (RE) set used. The yeast species of the ribose-seq libraries are also indicated. *S. cerevisiae* libraries derived from the same strains are grouped together by curly brackets. Thick, vertical, green lines separate data from the different yeast species. Vertical gray lines separate data obtained from different RNase H genotypes within each species. Each row shows results obtained for an rNMP (R in red) of base A, C, G, or U for each library. The actual percentages of A, C, G, and T bases present in nDNA of *S. cerevisiae* are shown to the left of the top heatmap, and are also indicated in Supplementary Data 2A. The corresponding base percentages for nDNA of *S. paradoxus* and *S. pombe* are indicated in Supplementary Data 2B and 2C, respectively. The bar to the right shows how different frequency values are represented as different colors: black for 0.25; black to yellow for 0.25 to 0.5–1, and black to light blue for 0.25 to 0. **b** Formulas used to calculate the frequency and the normalized frequency values of the nuclear heatmaps in **a**, see explanations in Methods.

abundant rNMP sites from each mitochondrial and nuclear library and analyzed them using the MEME program (see Methods). Because the number of rNMPs at a particular site can vary depending on sequencing coverage, calculating the top 1% allowed us to compare the frequency of rNMPs at each site among each library independently of the sequencing coverage of the libraries. For comparison, we also selected the top 100 most abundant rNMP sites. The results of both analyses revealed specific consensus motifs of rNMP incorporation for these hotspot sites in mtDNA (Fig. 7a, Supplementary Fig. 7A) and in nDNA for *rnh201*-null libraries (Fig. 7b, Supplementary Fig. 7B). We found rG followed by rC to be the most prevalent in all mitochondrial libraries except for those from strain E134, which had rC followed by rA. Mitochondrial libraries from

strains with *rnh202-pip*, *rnh1*-null, *rnh201*-null, and *rnh201-RED* mutations had similar rNMP preference as wild-type cells of the same strains. For hotspots in mitochondrial libraries of *S. paradoxus*, all had rC/rG as most abundant rNMP, while for *S. pombe*, rG was dominant followed by rC. Strikingly, dA was dominant at the −1 position in all hotspot motifs for all mitochondrial libraries of *S. cerevisiae*, *S. paradoxus*, and *S. pombe*, as well as in the *S. cerevisiae* emRiboSeq libraries (Supplementary Fig. 8A, B). The MEME analysis covered three nucleotides upstream and three downstream from the rNMP site. Although less pronounced, dT was conserved at position −3 and +2 in most of the mitochondrial *S. cerevisiae* libraries, including the emRiboSeq libraries, as well as in those of *S. paradoxus* and *S. pombe*. The consensus motif that emerged from all the hotspot

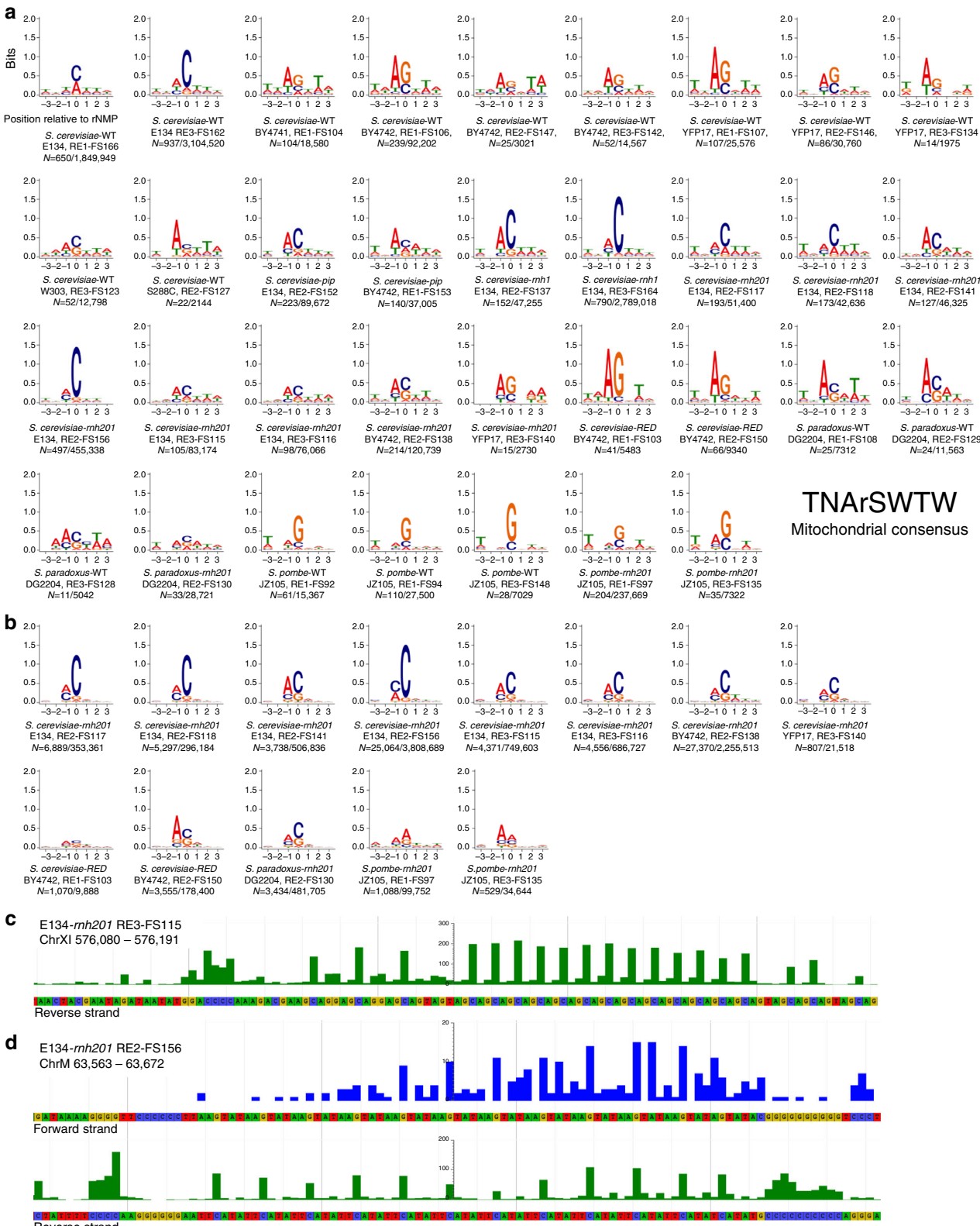

sites of the yeast mitochondrial libraries is TNArSWTW (Fig. 7a). Analysis of nDNA data from *rnh201*-null libraries revealed ArC as the dominant motif in all *rnh201*-null libraries of *S. cerevisiae*, including the *S. paradoxus rnh201*-null library. *S. pombe* showed ArA as the dominant motif in *rnh201*-null cells. No particular nucleotide further upstream or downstream from rC was conserved in the nDNA of these hotspot sites (Fig. 7b).

**Patterns of rNMPs in short-nucleotide repeats.** Concentration of rNMPs at short-nucleotide repeated tracts is a feature that emerged by browsing the genomic data of the ribose-seq libraries. We found a series of trinucleotide-repeated sequences that contain abundant rNMPs with a specific pattern. These are not artifacts of PCR because all reads in a given short-nucleotide repeated tract have different UMI, not just those reads that are at

**Fig. 7 Hotspot motifs with rNMPs in mtDNA and nDNA. a** Sequence motif plots for mitochondrial and **b** nuclear *rnh201*-null hotspots (top 1% of rNMP sites). Position 0 on the *x*-axis represents the site of rNMP incorporation, − and + positions represent upstream and downstream dNMPs, respectively. The *y*-axis shows the level of sequence conservation, represented in bits. The species, genotype, strain, restriction enzyme (RE) set, library name, and the number of rNMP sites are included below each plot. The consensus sequence for mitochondrial hotspots is shown: N any nucleotide, W weak (A or T), and S strong (G or C). **c** Genome browser snapshot showing an rC hotspot within the GAC-repeated sequence in *S. cerevisiae* nDNA at the locus chrXI:576134..576175 for ribose-seq library FS115 (*rnh201*-null). **d** Genome browser snapshot showing an rG hotspot within the TAAGTA-repeated sequence on the forward strand, and an rC hotspot in the complementary TACTTA-repeated sequence on the reverse strand in *S. cerevisiae* mtDNA at the locus chrM:63583..63651 for ribose-seq library FS156 (*rnh201*-null). All reads shown here have distinct UMI, meaning they do not represent PCR duplicates caused by slippage of DNA polymerase. Similarly, reads for other hotspot sites within short-nucleotide repeats are shown in Supplementary Fig. 9.

the same position within the repeated tract. In *S. cerevisiae*, for example, region chrXI:576134..576175 on the reverse strand for nuclear library FS115 (E134 *rnh201*-null) displayed multiple hotspots of rNMP incorporation at the C-nucleotide in the triplet GAC (Fig. 7c). The same pattern was reproducible at the same locus in FS116 (E134 *rnh201*-null) and FS140 (YFP17 *rnh201*-null). Another interesting example is locus chrM:63583..63651 in FS156, also seen in FS141, FS138, FS162, and FS166, that has rNMPs at the G-nucleotide of TAAGTA-repeated sequence on the forward strand and at the C-nucleotide on the reverse strand in TACTTA-repeated sequence (Fig. 7d). Interestingly, this pattern was also evident in emRiboSeq libraries SRR1734980 (Supplementary Fig. 8C) and SRR173982. A series of similar patterns with ArC in trinucleotide, dinucleotide, or short-nucleotide repeat tracts were found in other loci in the *S. cerevisiae* genome (Supplementary Fig. 9A–F).

## Discussion

We report a genome-wide analysis of rNMP sites in mtDNA and nDNA of wild type and RNase H mutants of different strains of *S. cerevisiae*, *S. paradoxus*, and *S. pombe*. Our results, observed consistently across multiple genotypes and/or strains, show that rNMPs are found at preferential sites in DNA, rather than being randomly distributed along the mitochondrial and nuclear genomes. A remarkable feature across *S. cerevisiae* and *S. paradoxus* and in most part conserved in *S. pombe* is that the deoxyribonucleotide immediately upstream from the rNMPs embedded in DNA is found to have the strongest impact on rNMP distribution in DNA, compared to the nucleotide immediately downstream or those further up or downstream. This feature may reflect an accommodation mechanism of yeast replicative DNA polymerases that is favored by specific sequence contexts. Exonucleolytic proofreading activity on rNMPs by DNA polymerases (Pols) is absent or weak. No proofreading was detected for human mitochondrial Pol γ[24] or yeast Pol δ, while only limited proofreading activity was shown for Pol ε on rU[3,21,36]. Studies using X-ray crystal structures of human DNA polymerase μ, which is very promiscuous in rNMP incorporation, showed that the enzyme can accommodate rNMPs almost as well as dNMPs in its active site, with no alteration of protein structure[37]. While yeast nuclear Pol δ and ε, and mitochondrial Pol γ have stronger sugar discrimination capacity than Pol μ (see refs. [16,38] and references therein), it is possible that specific sequence contexts provide for sufficient accommodation of the rNMPs in the polymerase active site, allowing rNMP incorporation in DNA by these replicative enzymes.

We found that the overall distribution and dinucleotide patterns of rNMPs in *S. cerevisiae* and *S. paradoxus* are similar. The rNMP patterns in *S. pombe* have more similarity with the budding yeasts at the nuclear than at the mitochondrial level. Mitochondrial rC and rG were found in C/G-rich regions in the budding yeasts but not in *S. pombe* (Fig. 3). rC and rG were often found downstream of dA in nDNA of all the three species, while

this was evident only in mtDNA of the budding yeasts. In fact, the dinucleotide patterns show clear similarities at the nuclear level across the three yeast species (Fig. 4b). A major common feature across the three yeast species studied here is the strong similarity of the rNMP patterns of mtDNA between wild type and *rnh201*-null cells of the same species and strains. Not only did we confirm the lack of activity of RNase H2 in *S. cerevisiae* mitochondria, but we also found that this applies to mtDNA of *S. paradoxus* and *S. pombe*. Interestingly, despite many conserved features of rNMP patterns across the yeast species, we revealed variation among yeast strains of the same species. Particularly, within the six different strains of *S. cerevisiae* studied, strain E134 displayed marked preference for rC vs. rG in mtDNA (Fig. 2a). Although different yeast strains may have DNA sequence polymorphisms, it is unlikely these prominently alter the A, C, G, and T content of the genome. Lower ratio for rGTP/dGTP in E134 compared to BY4742 strain may account for such preference. We found more consistent rNMP distribution in nDNA among all the strains for wild-type RNase H2 cells of *S. cerevisiae*, and similarly for *rnh201*-null cells (Fig. 6a). Another common feature among the three yeast species is the underrepresentation of rU in mtDNA of any genotype, and in nDNA of *rnh201*-null cells. While the low level of rU can be explained by general high concentration of dTTP in the nucleotide pools[3,23] with rUTP/dTTP being the lowest rNTP/dNTP ratio, which we observed both in *S. cerevisiae* and *S. pombe* strains (Supplementary Fig. 1), potential activity of topoisomerase I on sequences with rU[12,39], and some proofreading activity for Pol ε on rU[21], its incorporation in mtDNA and nDNA of *rnh201*-null cells is not random. In fact, we show that rU is found in most cases after dC, and this feature is highly conserved across *S. cerevisiae*, *S. paradoxus,* and *S. pombe* (Fig. 4).

Wild-type RNase H2 cells have active RER to remove incorporated rNMPs from nDNA. Therefore, we expected to detect transiently incorporated rNMPs, not yet removed by RER, or rNMPs that escaped RER removal. Interestingly, we found rG to be present below the proportion of dGMP in all nuclear wild-type libraries of *S. cerevisiae*, *S. paradoxus*, and *S. pombe*. This was also noted for *S. cerevisiae* strain W1588-4C[23], derivative of W303. It was shown that lack of RNase H2 functionality does not alter nucleotide pools in *S. cerevisiae* cells[23]. Therefore, the fact that wild-type RNase H2 cells have low rG and equal absolute amount of rA, rC, and rU in nDNA, with an overall rNMP distribution that is significantly different from that observed in *rnh201*-null cells (Figs. 2a, b and 6a, b), may indicate that yeast RNase H2 cleaves rNMPs in yeast DNA with differential efficiency and may have preference for rG, as preliminary data showed, using protein extracts from yeast, HeLa cells, and *Escherichia coli* cells[40]. A more recent study, exploiting microarray analysis of thousands of rNMP-containing DNA hairpins of different sequence, demonstrated clear cleavage preference of *E. coli* RNase HII at and around the cleavage site[41]. It is also possible that other mechanisms to remove rNMPs from DNA may be more active in *rnh201*-null than in wild-type cells, like topoisomerase I[12,39],

which may have preferred genomic targets, or both cases may apply.

The presence of rG was also not random. Like rC, rG was most often preceded by dA both in *S. cerevisiae* and *S. paradoxus* mitochondrial libraries and in all nuclear libraries across all the three yeast species for all strains and genotypes examined in this study. Strikingly, ArC and ArG were dominant motifs in all hotspots, as well as in trinucleotide, dinucleotide, and other short-nucleotide repeats across nuclear and mtDNA. Interestingly, we found enrichment of rNMPs in short-nucleotide repeated sequences. The observed patterns of rNMPs demonstrate a strong propensity for rNMP incorporation when nucleotides with base C or G are in di, tri, or other short-nucleotide repeats. Remarkably, in all repeats that we analyzed except one (9/10), the rCs and rGs were preceded by dA (Fig. 7c, d, Supplementary Figs. 8C and 9). Thus, if the same dinucleotide-sequence context with AC or AG is repeated within a given region of the genome, like in trinucleotide or dinucleotide repeated tracts containing such dinucleotides, any AC or AG site within the repeated tracts may be equally prone to rNMP incorporation, explaining the observed patterns. It is tempting to speculate that these rNMP sites at short-nucleotide repeats can contribute to the instability of these regions, possibly driving mechanisms of repeat expansion/contraction.

In conclusion, via mapping and genome-wide analysis of many ribose-seq libraries across three yeast species with wild type and mutant RNase H2 genotypes, which consistently generated reproducible results, we reveal new biological features and a fundamental rule shaping the patterns of rNMP incorporation. Not only is genomic rNMP incorporation far from random, but it is also driven by the sequence context, particularly by the sequence of the nucleotide immediately upstream of the site of incorporation. Potentially, such sequence context-driven mechanism of rNMP incorporation reflects a physiological function of rNMPs in DNA. More broadly, our work provides robust tools and a model framework to further study biological, chemical, and structural aspects of genomic rNMP incorporation from yeast throughout all kingdoms of life.

## Methods

**Yeast strains**. All the yeast strains used in this study are presented in Supplementary Table 1. We used haploid *Saccharomyces cerevisiae* strains from different backgrounds: E134, BY4741, BY4742, YFP17, W303, and S288C. In addition to *S. cerevisiae*, we also used *Saccharomyces paradoxus* (DG2204) and *Schizosaccharomyces pombe* (JZ105). Standard genetic and molecular biology methods were used for yeast growth, gene disruption, isolation of mutants, yeast marker selection, yeast genome engineering, yeast colony PCR, and sequence analysis of yeast DNA[42–44]. All *RNH201* deletion strains were made by replacing *RNH201* via transformation with a PCR product containing *hygMX4* or *kanMX4* cassette flanked by 50 nucleotides of sequence homologous to regions upstream and downstream of *RNH201* ORF. KK-172 was made from KK-44 by replacement of *RNH1* with the *kanMX4* cassette. Yeast strains SB-285 and SB-286 were derived from KK-44 by using the delitto perfetto method[44] and then by popping out the CORE cassette by a pair of oligonucleotides, primers 202PIP.F and 202PIP.R to mutate the PCNA-interacting peptide box, which is present in Rnh202 to make *rnh202*-FF346,347A[18,34]. SB-311 was made from KK-2 by using the delitto perfetto method to generate *rnh201*-P45D and *rnh201*-Y219A by using primers RNH P45D.60 and RNH Y219A.60 to result in an RNase H2 ribonucleotide-excision defective (RED)[34] mutant. All mutations were confirmed by sequence analysis of PCR products obtained from amplification of a DNA region surrounding the specific mutation. Primers used for strain construction are listed in Supplementary Table 2.

**Properties of ribose-seq**. The ribose-seq technique allows to build libraries of rNMP sites that are present in any DNA source of interest[21,29]. The key enzyme of ribose-seq is *Arabidopsis thaliana* tRNA Ligase (AtRNL). AtRNL recognizes the 2′,3′ cyclic phosphate end of a single-stranded (ss) genomic DNA fragment terminated with an rNMP, generated upon treatment of double-stranded DNA by alkali. AtRNL directly ligates the rNMP-terminated ssDNA to the 5′-phosphate end of the same ssDNA fragment, to which an adaptor sequence has been attached before the alkali treatment[21,29]. Such ssDNA circles, each containing one rNMP

next to the adaptor, constitute the rNMP library for a given DNA sample. Ribose-seq cannot capture primers of Okazaki fragments because these do not have an adaptor ligated at their 5′ end and are degraded upon alkali treatment. Moreover, thanks to the specificity of the enzyme activity, AtRNL cannot capture abasic sites or DNA sequences upstream or downstream from DNA breaks[21]. Thus, cell cycle stage and/or integrity of DNA do not generate false positives in the preparation of the ribose-seq libraries from the DNA samples of choice.

**Choice of yeast backgrounds for ribose-seq library preparation**. We built 34 ribose-seq libraries of yeast mtDNA and 25 libraries of yeast nDNA. These include 25 mitochondrial and 18 nuclear libraries derived from *S. cerevisiae* of six different strain backgrounds E134, BY4741, BY4742, YFP17, W303, and S288C, four mitochondrial and three nuclear from *S. paradoxus* of strain DG2204, and five mitochondrial and four nuclear from *S. pombe* of strain JZ105 (Supplementary Table 1). For *S. cerevisiae*, we utilized some of the most commonly used haploid yeast laboratory strains. Strain S288C was used in the systematic sequencing project of *S. cerevisiae*[45]; strains BY4741 and BY4742 both are derivatives of S288C, were used in the *S. cerevisiae* gene disruption project, and have opposite mating type[46]; W303 is another common yeast laboratory strain more distantly related to S288C[47]; YFP17 is a derivative of DBY745 (*MATα ura3-52 leu2-3 leu2-112 adel-100*)[48]; and E134 is a derivative of CG379 (ref. [49]). The ribose-seq libraries derive from wild-type RNase H2 or null *rnh201* cells of all three yeast species, and also from *S. cerevisiae* cells containing the *RED* mutations of *rnh201* (P45D-Y219A), which block RNase H2 activity on single rNMPs in DNA but allow cleavage at long RNA/DNA hybrids[34], the *pip* mutation in *rnh202* (FF346,347AA), which impedes interaction with PCNA[50], or a null-*rnh1* allele (Supplementary Table 1). For wild-type cells in the different species, and for RNase H-mutant cells in *S. cerevisiae*, we constructed two or more ribose-seq libraries using two or three different sets of restriction enzymes (RE1: *Dra*I, *Eco*RV, and *Ssp*I; RE2: *Alu*I, *Dra*I, *Eco*RV, and *Ssp*I; and RE3: *Rsa*I, and *Hae*III). This strategy allowed us (i) to verify that the conclusions taken from our analyses of ribose-seq data are not influenced by a particular set of restriction enzymes used to fragment the DNA extracted from the different yeast species and genotypes, and (ii) to further confirm reproducibility of the results. All ribose-seq libraries have a specific barcode within the sequence of the Unique Molecular Identifier (UMI) to distinguish the libraries from each other in the sequencing run and eliminate PCR duplicates (see ref. [30], Methods, Supplementary Table 2).

**Ribose-seq library preparation**. Libraries were prepared using the ribose-seq method with some modifications[21,29]. Specifically, we optimized the ribose-seq protocol by (i) redesigning the molecular barcode-containing adaptor, making it shorter and removing overlapping sequences, (ii) fragmenting the genome of interest in smaller fragments (~450 bp); (iii) performing two rounds of PCR and overall reducing the PCR cycle number; (iv) cutting and purifying a specific size-range of the ribose-seq library from the non-denaturing gel to eliminate any primer dimers formed during PCR and any long products that are not proficient for sequencing[29]. All the commercial enzymes utilized in the ribose-seq protocol were used according to the manufacturer's instructions. *S. cerevisiae* cells were cultured in liquid rich medium (150 mL of a 250 mL glass flask) containing yeast extract, peptone, and 2% (wt/vol) dextrose (YPD) for 2 days at 30 °C with shaking to reach stationary phase with a density of ~$10^8$ cells/mL. Genomic DNA was extracted using Qiagen Genomic DNA protocol "Preparation of Yeast Samples". Successively, 40 μg of yeast genomic DNA were fragmented using restriction enzymes to produce blunt-ended fragments with an average size of 450 base pairs (bp) in length. Multiple sets of restriction enzymes were used for different library preparation, as shown in Supplementary Data 1 and 4. The different combinations used were (i) RE1: *Dra*I, *Eco*RV, and *Ssp*I; (ii) RE2: *Alu*I, *Dra*I, *Eco*RV, and *Ssp*I; and (iii) RE3: *Rsa*I, and *Hae*III. Following restriction digestion, the fragmented DNA was purified by spin column (Qiagen). The fragments were tailed with dATP (Sigma Aldrich) by using Klenow Fragment (3′→5′ exo-) (NEB) for 30 min at 37 °C and purified by using spin column. Following dA-tailing and purification, the DNA fragments were annealed with a partially double-stranded adaptor (Adaptor. L1 or Adaptor.L2 with Adaptor.S, Supplementary Table 2) by using T4 DNA ligase (NEB) incubating overnight at 15 °C. Following overnight ligation, the products were purified using RNAClean XP beads (Beckman Coulter). The annealed fragments were treated with 0.3 M NaOH for 2 h at 55 °C to denature the DNA strands, and to cleave at the rNMP sites resulting in 2′,3′-cyclic phosphate and 2′-phosphate termini. This was followed with neutralization using 0.3 M HCl and purification using RNAClean XP beads. All the successive purification steps were performed using RNAClean XP beads. The single-stranded ssDNA fragments were incubated with 1 μM *Arabidopsis thaliana* tRNA ligase (AtRNL), 50 mM Tris-HCl pH 7.5, 40 mM NaCl, 5 mM MgCl$_2$, 1 mM DTT, and 300 μM ATP in a volume of 20 μL for 1 h at 30 °C, followed by purification. AtRNL aids in ligating the 2′-phosphate ends of rNMP-terminated ssDNA fragment to its opposite 5′-phosphate end, which results in a circular ssDNA. Due to the efficient removal of rNMPs by RNase H2, nuclear libraries of wild-type *RNH201*, *rnh202-pip*, and *rnh1*-null cells generally had a much lower number of reads compared to the mitochondrial libraries of the same cells, and thus had a higher number of background reads that originated from the capture of restriction enzyme ends likely by residual activity of T4 DNA ligase. These background reads were identified computationally and were found to

constitute less than 5% of the total reads in the mitochondrial and less than 7.5% nuclear *rnh201*-null and *rnh201-RED* libraries. To determine whether the background reads could influence our results and conclusions, we subtracted the background reads from the total reads. When the background reads were <12% of the total reads, the results after background subtraction were found to be the same as those without subtraction. Therefore, for our analyses, we selected only those nuclear libraries for which the background was <12%, so that our results and conclusions are not biased (see also below). The fragments were then treated with T5 Exonuclease (NEB) 50 units in 50 μL volume for 1 h 30 min at 37 °C to degrade the unligated ssDNA fragments. After purification, the circular fragments were incubated with 1 μM 2′-phosphotransferase (Tpt1), 20 mM Tris-HCl pH 7.5, 5 mM MgCl₂, 0.1 mM DTT, 0.4% Triton X-100, and 10 mM NAD⁺ in a volume of 40 μL for 1 h at 30 °C to remove the 2′-phosphate present at the ligation junction. After Tpt1 treatment and purification, the circular fragments were PCR-amplified using two rounds of amplifications to result in ribose-seq library: both PCR rounds begin with an initial denaturation at 98 °C for 30 s. Then denaturation at 98 °C for 10 s, primer annealing at 65 °C for 30 s, and DNA extension at 72 °C for 30 s are performed. These three steps are repeated for 6–15 cycles in the first PCR round, and for 7–13 cycles in the second PCR round depending on the concentration of the circular ssDNAs containing the rNMPs. Successively, there is a final extension reaction at 72 °C for 2 min for both PCRs.

A first round of PCR was performed to amplify and introduce the sequences of Illumina TruSeq CD Index primers. The primers (PCR.1 and PCR.2) used for the first round were the same for all libraries. A second round of PCR was performed to attach specific indexes i7 and i5 for each library. The sequences of PCR primers and indexes can be found in Supplementary Table 2. PCR round 1 and 2 were performed using Q5-High Fidelity polymerase (NEB) for 6 and 7 cycles, respectively (unless specified otherwise). Following the PCR cycles, the ribose-seq library was loaded on a 6% non-denaturing polyacrylamide gel and stained using 1× SYBR Gold (Life Technologies) for 40–45 min. As shown in Koh et al.[21], in control experiments for the optimized ribose-seq protocol, we found that exclusion of either AtRNL or alkali treatment prevented library formation (Supplementary Fig. 10). Fragments between 200 and 700 bp were cut and gel purified using the crush and soak method[51]. The resulting ribose-seq libraries were mixed at equimolar concentrations and normalized to 1.5 nM. The libraries were sequenced on an Illumina MiniSeq in the Molecular Evolution Core Facility at the Georgia Institute of Technology.

**dNTP and rNTP measurements**. Yeast cell lysate were appropriately prepared to extract the dNTPs and rNTPs using an established protocol[52] with some modifications. Yeast cells were grown as described above in YPD medium for 2 d at 30 °C with shaking to reach the stationary phase with a density of ~10⁸ cells/mL. Cell were then harvested, washed with DI water, and resuspended in a solution of 1 M sorbitol, 100 mM EDTA, 14 mM B-mercaptoethanol, and 1 mg of Zymolase and incubated at 37 °C for 2 h. The mixture was then spun down and the pellet of cells were washed two times with a Phosphate Buffer Saline solution. The pellet was then resuspended in 65% methanol and mixed by pipetting. The mixture was heated at 95 °C for 3 min and then placed on ice for 1 min. The cells were spun at 16,000 × *g* for 3 min. The supernatant was transferred to a 30 kDa column where it is spun for 30 min at 18,000 × *g*. The flow through was lyophilized and stored at −80 °C. To quantify the intracellular dNTPs and rNTPs, an ion pair chromatography-tandem mass spectrometry method[53] was applied, with modifications. Chromatographic separation and detection were performed on a Vanquish Flex system (Thermo Scientific) coupled with a TSQ Quantiva triple quadrupole mass spectrometer (Thermo Scientific). Analytes were separated using a Kinetex EVO-C18 column (100 × 2.1 mm, 2.6 μm) (Phenomenex) at a flow rate of 250 μL/min. The mobile phase A consisted of 2 mM of ammonium phosphate monobasic and 3 mM of hexylamine in water and the mobile phase B consisted of acetonitrile. The LC gradient increased from 10% to 35% of mobile phase B in 5 min, and then returned to the initial condition. Selected reaction monitoring in both positive and negative modes (spray voltage: 3200 V (pos) or 2500 V (neg); sheath gas: 35 Arb; auxiliary gas: 20 Arb; ion transfer tube temperature: 350 °C; vaporizer temperature: 380 °C) was used to detect the targets: dATP (492→136, pos), dGTP (508→152, pos), dCTP (466→158.9, neg), TTP (481→158.9, neg), ATP (508→136, pos), GTP (524→152, pos), CTP (482→158.9, neg), UTP (483→158.9, neg). Extracted samples were reconstituted in 100 μL of mobile phase A. After centrifuging at 13,800 × *g* for 10 min, 40 μL of supernatant was mixed with 10 μL of 13C and 15N labeled dNTPs and rNTPs as internal standards, and then subjected to analysis. Data were collected and processed by Thermo Xcalibur 3.0 software. Calibration curves were generated from standards by serial dilutions in mobile phase A (dATP and dGTP 0.1–400 nM, dCTP and TTP 0.2–400 nM, rNTPs 1–4000 nM). The calibration curves had *r*2 value greater than 0.99. All the chemicals and standards are analytical grade or higher and were obtained commercially from Sigma Aldrich. Nucleotides were at least 98% pure.

**Processing and alignment of sequencing reads**. For the ribose-seq libraries, the sequencing reads consist of an eight-nucleotide UMI, a three-nucleotide molecular barcode, the tagged nucleotide (the nucleotide tagged during

ribose-seq from which the position of the rNMP is determined), and the sequence directly downstream from the tagged nucleotide. The UMI corresponds to sequencing cycles 1–6 and 10–11, the molecular barcode corresponds to cycles 7–9, the tagged nucleotide corresponds to cycle 12, and the tagged nucleotide's downstream sequence corresponds to cycles 13+ of the raw FASTQ sequences. The rNMP is the reverse complement of the tagged nucleotide. Before aligning the sequencing reads to the reference genome, the reads were trimmed based on sequencing quality and custom ribose-seq adaptor sequence using cutadapt 1.16 (-q 15 -m 62 -a "AGTTGCGACACGGATCTATCA"). In addition, to ensure accurate alignment to the reference genome, reads containing fewer than 50 bases of genomic DNA (those bases located downstream from the tagged nucleotide) after trimming were discarded. Following quality control, the Alignment and Coordinate Modules of the Ribose-Map toolkit were used to process and analyze the reads[30]. The Alignment Module de-multiplexed the trimmed reads by the appropriate molecular barcode, aligned the reads to the reference genome using Bowtie 2, and de-duplicated the aligned reads using UMI-tools. Based on the alignment results, the Coordinate Module filtered the reads to retain only those with a mapping quality score of at least 30 (probability of misalignment <0.001) and calculated the chromosomal coordinates and per-nucleotide counts of rNMPs. All ribose-seq libraries were then checked for background noise of restriction enzyme reads. We counted the number of reads ending with a restriction enzyme cut site, which is expected not to be generated by ribonucleotides incorporation. Some reads captured the dAMP, which is added by dA-tailing at the restriction cut site. We summed up such background reads and calculated the percentage of background noise. All mitochondrial libraries (34/34) had very low background (0.04–4.85%). Majority of the nuclear libraries (25/33) had background <12% (0.02–11.74%), and these were studied. To allow comparison between sequencing libraries of different read depth, the per-nucleotide coverage was calculated by normalizing raw rNMP counts to counts per hundred. For the emRiboSeq libraries, we downloaded libraries SRR1734967, SRR1734969, SRR1734972, SRR1734980, and SRR1734982 from NCBI's SRA using the SRA toolkit and obtained the chromosomal coordinates of rNMP sites using the Alignment and Coordinate Module of Ribose-Map. The FASTQ files and configuration files used as input into the Alignment and Coordinate Modules are available as Supplementary Material at NCB online.

**Nucleotide sequence context of embedded rNMPs**. Using the Sequence Module of Ribose-Map, the frequencies of the nucleotides at rNMP sites and 100 nucleotides upstream and downstream from those sites were calculated for the nuclear and mitochondrial genomes. The Sequence Module normalizes the nucleotide frequencies to the frequencies of the corresponding reference genome. To normalize the nucleotide frequencies, the number of each type of nucleotide (A, C, G, U/T) present in the ribose-seq data was counted and divided by the total number of nucleotides at a given position to yield the raw proportion. In addition, the number of each type of nucleotide (A, C, G, T) present in the reference genome was counted and divided by the total number of nucleotides in the reference genome to yield the reference proportion. Then, the raw proportions were divided by the corresponding reference proportions to yield the normalized nucleotide frequencies.

**Data presentation**. Bar graphs representing the percentage of rNMPs were made using GraphPad Prism 8 (GraphPad Software). The nucleotide sequence context plots were created using the ggplot2 R package. Consensus sequences around rNMP sites were identified using Multiple Em for Motif Elicitation (MEME)[54] and plotted using ggseqlogo[55].

**Heatmaps**. To generate the mononucleotide heatmaps, in every mitochondrial and nuclear ribose-seq library, the number of each type of rNMP ($R_N$: $R_A$, $R_C$, $R_G$, or $R_U$) was counted and divided by the total number of rNMPs to yield the proportion $R_{N_{R,\,raw}}$:

$$R_{N_{R,\,raw}} = \frac{R_N}{R_A + R_C + R_G + R_U}.$$

Then, each raw count of rNMPs ($R_N$) was divided by the corresponding deoxy-mononucleotide frequency of the reference genome ($N_N$, see percentage rows in Supplementary Data 2) to yield the probability of rNMP incorporation $P_{N_R}$, and were normalized to generate the normalized proportion $R_{N_{R,\,norm}}$. These data were used in the normalized mononucleotide heatmaps:

$$P_{R_N} = \frac{R_N}{N_N} \quad R_{N_{R,\,norm}} = \frac{P_{R_N}}{P_{R_A} + P_{R_C} + P_{R_G} + P_{R_U}}.$$

Similarly, to generate the normalized dinucleotide heatmaps, each raw count of dinucleotides with an rNMP along with a deoxyribonucleotide at position −1, −2, −3, −4, or −100 relative to the rNMP ($R_{NR}$), or at position +1, +2, +3, +4, or +100 relative to the rNMP ($R_{RN}$) was divided by the corresponding deoxy-dinucleotide frequency of the reference genome ($N_{NN}$, see percentage rows in Supplementary Data 2) to yield the probability of *NR* and *RN* dinucleotide incorporation $P_{R_{NR}}$ and $P_{R_{RN}}$. Next, these proportions were normalized keeping

fixed the deoxyribonucleotide base in the position of the rNMP to generate the normalized proportion $R_{R_{NR, norm}}$ or $R_{R_{RN, norm}}$:

$$P_{R_{NR}} = \frac{R_{NR}}{N_{NN}} \quad R_{R_{NR, norm}} = \frac{P_{R_{NR}}}{P_{R_{AR}} + P_{R_{CR}} + P_{R_{GR}} + P_{R_{TR}}},$$

$$P_{R_{RN}} = \frac{R_{RN}}{N_{NN}} \quad R_{R_{RN, norm}} = \frac{P_{R_{RN}}}{P_{R_{RA}} + P_{R_{RC}} + P_{R_{RG}} + P_{R_{RT}}}.$$

All species-specific background frequencies of mononucleotides and dinucleotides are shown in Supplementary Data 2.

**Statistical analysis for heatmaps**. To compare frequency results of heatmap data obtained for each rNMP, or dinucleotide pair containing an rNMP, with those obtained with all other rNMPs, or dinucleotide pairs containing an rNMP, within a specific genotype of mitochondrial or nuclear libraries for a given yeast species, we used the two-sided Mann–Whitney $U$ test. Each pair of mononucleotides, or dinucleotide pairs containing the same rNMP, was tested to determine whether its frequency was significantly greater or smaller than that of the other samples. These data are shown in Supplementary Data 3.

**Genome browser and hotspots**. BedGraph files were generated using the Distribution Module of Ribose-Map and then visualized using the JBrowse genome browser[56] and then visualized using the JBrowse genome browser. Top 1% and top 100 most abundant rNMP sites for the ribose-seq and emRiboSeq libraries were calculated based on the BED files created by the Coordinate Module of Ribose-Map. For the analysis of short-nucleotide repeat tracts, we only considered reads that were longer than the repeated regions to ensure accuracy of our findings.

**Reporting summary**. Further information on research design is available in the Nature Research Reporting Summary linked to this article.

## Data availability
The authors declare that the data supporting the findings of this study are available within the paper and its supplementary information files. The ribose-seq datasets generated during the current study are available in NCBI's Sequence Read Archive via BioProject "PRJNA613920". The emRiboSeq datasets analyzed during the current study are available in NCBI's Gene Expression Omnibus via accession number "GSE64521". The files of Supplementary Data 1, 4 and 5 contain raw data. All data are available from the authors upon reasonable request.

## Code Availability
The Ribose-Map bioinformatics toolkit is available for download at GitHub (https://github.com/agombolay/ribose-map). Customized python3 scripts for background subtraction are available on GitHub under GPLv3.0 license (https://github.com/xph9876/ArtificialRiboseDetection). Customized python3 scripts for heatmaps are also available on GitHub under GPLv3.0 license (https://github.com/xph9876/RibosePreferenceAnalysis).

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

## Acknowledgements

We thank Y. Chernoff for yeast strains BY4741, W303, and S288C; S. Biliya and N. Djeddar for high-throughput sequencing and sequencing advices; T. Hilley for help setting up the genome browser, the Partnership for an Advanced Computing Environment (PACE) at the Georgia Institute of Technology for their research cyberinfrastructure resources and services; M. W. Germann for discussion of results; P. McGrath and S. Marsili for critically reading the manuscript; and all members of the Storici laboratory for assistance and feedback on this study. We acknowledge funding from the National Institutes of Health, NIH AI136581 (to B.K.), AI150451 (to B.K.), MH116695 (to R.F.S.), R01ES026243 (to F.S.), the Parker H. Petit Institute for Bioengineering and Bioscience at Georgia Institute of Technology 12456H2 (to F.S.), and the Howard Hughes Medical Institute Faculty Scholar grant 55108574 (to F.S.) for supporting this work.

## Author contributions

F.S. together with S.B., A.L.G., T.Y., P.X., and K.D.K. conceived the project and designed experiments. F.S. wrote the manuscript with help from S.B., A.L.G., T.Y., and P.X. S.B. and T.Y. performed most of the experiments with help from G.N., H.K., W.M.M.E.-S., and A.V.B. A.L.G. and P.X. performed all the bioinformatics analysis of the data with guidance from F.O.V. S.T., N.E.B., R.F.S., and B.K. performed the measurements of dNTPs and rNTPs. All authors commented on and approved the manuscript.

## Competing interests
The authors declare no competing interests.
