## [Peer Review File · Nature Communications]

Reviewers' comments:

Reviewer #1 (Remarks to the Author):

In this manuscript, the authors have used a single method, Ribose-seq, on DNA from *S. cerevisiae*, *S. paradoxus* and *S. pombe*, to identify the base identity of incorporated ribonucleotides, the nucleotide-sequence context of incorporated ribonucleotides and highlight patterns of ribonucleotide in repeats in each strain. They also investigated if these parameters changed upon deletion of *rnh1*, *rnh201* or in strains with a *pip* mutant or RED-mutant in *S. cerevisiae*.

Several studies have already highlighted the base identity of incorporated ribonucleotides in yeast (Wanrooij et al, Clausen et al) and in humans (Berglund et al, CF Moss et al), however it has not been published that dA is preferred upstream of an incorporated ribonucleotides. In total they analyzed 34 different strains, but unfortunately there is no replicate of individual strains, and it is therefore not easy to judge if the observed variation is due to different strains or due to technical variation. Figure 1a and b and Figure 4a and b have been published before for *S. cerevisiae*. The manuscript does not provide much further biological function of ribonucleotides in DNA and I do not believe this study add enough news value to the already published work.

Minor comments:

Abstract, There are certainly significant conservation of ribonucleotide incorporation, but how often are specific sites imprinted with ribonucleotides?

Page 3, What are the substantial modifications to the original Ribose-seq and how does that change the data?

Page 3, Supplementary table 3-5 is missing.

Page 11, The authors analyze so called hotspots of ribonucleotide incorporation, these hot spots could be generated during PCR amplification so it would be important to show that these are unique hotspots and not only PCR products which are amplified more efficient.

Page 12, How did you analyze the libraries in the genome browser?

Page 3, Supplementary: The ribose-seq protocol lacks volumes and units used for each enzyme, nucleotide concentrations, adapter concentrations and the PCR program used.

Figure 1d-i, labels with each strain is missing.

Figure 2: labels with each strain is missing.

Figure 3: why are there fewer strains used in panel b, compared to a?

Reviewer #2 (Remarks to the Author):

In this study, Balachander et al. performed systematic analysis of ribonucleotide identity and their sequence context in the genome using Ribose-seq approach previously developed by the Storici group. Their analysis includes two *Saccharomyces* species and the *Schizosaccharomyces pombe*. Consistent with previous reports, they confirm preferential incorporation/retention of certain bases

over others, eg. preference for rC in RNase H2-deficient *Saccharomyces*. They uncovered that the ribonucleotide incorporation is influenced by immediate sequence upstream of rNMP, eg. dA is preferred upstream of rC and rG. These are interesting observations. However, it is not immediately clear what the implication may be for the biology of ribonucleotide incorporation. Moreover, in my opinion, validation of the overall approach has yet to be established.

Specific concerns and suggestions:

1. Enzymatic reactions involving nucleotide sequences have potential biases. Since the authors are examining arguably small to moderate differences in rNMP abundance and sequence context preferences, the intrinsic bias of the library preparation, if true, can significantly alter the results. The relative abundance of rNMP has been reported in some of the yeast species using alternative rNMP mapping techniques. This allows for comparison of observations using different approaches. A previous study using the PU-seq approach (Daigaku et al. 2015) reports that rA and rU are the most abundant rNMPs in RNase H2 deficient-*S. pombe* cells (Figure S3). In contrast, the current study identified rA as the most abundant and rU as the least abundant. The sequence context preference report is also somewhat different between the two studies. This apparent discrepancy is perhaps due to the difference in the intrinsic biases in different techniques. Ligation reactions often has been documented for several ligation-based Illumina library preparation methods. Thus, without further validation, it is unclear whether the sequence preferences, which is the central focus in this study, is due to bias in ribonucleotide incorporation or bias of the approach, or perhaps more likely, a combination of both.

2. In the Supplementary Methods, it was stated that "AtrNL cannot capture abasic sites or ... DNA breaks". However, in later sections, the authors acknowledged that reads from restriction sites contributed to background noise, apparently in some cases, up to or over 12% of the total reads. This contradicts the statement about the specificity of AtrNL. If the restriction sites contribute to the noise, then other random nicks and breaks can also be captured. One way to address the issue is perhaps to construct control libraries without alkali treatment.

3. In the Introduction, the authors stated that the current ribose-seq technique has been substantially improved relative to the original version and referred to a methods paper that is in press. It would be beneficial to general audience if the authors could highlight the modifications they made and how they improved the quality of the data presented.

4. Yeast culture conditions (media and potential supplements and density of culture (log phase vs stationary)) may influence ribonucleotides identified in the cell population. The authors need to be more specific on these conditions in the Methods section.

5. It is clear from the data that the rNMP compositions are different in different species, particularly between *Saccharomyces* and *Schizosaccharomyces*. It would be helpful to understand whether this difference is dictated by the difference in the rNTP and dNTP pools. The authors could consider measuring nucleotide pools in these strains.

6. In Figures 1b and 4b, after normalization to background, the expected mononucleotide frequency should be 25% for each. Thus, it is misleading to label the background frequency next to the normalized data. It may be more intuitive to move these numbers next to the upper panel. It also seems inappropriate to only indicate the *S. cerevisiae* genome nucleotide composition. The same is true for the dinucleotide heatmaps.

7. "These results suggest that rGs tend to be hotspot sites of rNMP incorporation in mitochondrial DNA of wild-type RNase H cells." I think the authors meant "rC" in this sentence.

8. Figure 1d-i. It is unclear what the y-axis means. The frequency exceeds 1 at some positions.

9. The criteria for “common rNMP sites” need to be better defined. Is the lack of “common sites” in the nuclear genome libraries due to the lower sequence coverage in comparison with mitochondria libraries?

10. The authors selected top 100 rNMP sites for motif analysis. This seems arbitrary. Why not use all identified rNMP sites?

11. In Figures 5 c, d, the authors show examples of individual rNMP hotspots. It would be important to show that these potential hotspots are not due to certain artifacts not related to rNMPs. Thus, preparing libraries without alkali treatment seems necessary.

12. “Interestingly, we found rG to be present below the standard level of G in all nuclear wild-type libraries of *S. cerevisiae*, *S. paradoxus* and *S. pombe*.” Define “standard level”.

Reviewer #3 (Remarks to the Author):

The Storici lab developed one of three techniques that can be used to map the identities/positions of rNMPs embedded in genomic DNA and ribose-seq is the only technique that eliminates the background noise of DNA damage. In the current study, this technique is used to map rNMPs in multiple *S. cerevisiae* strains, an *S. paradoxus* strain and an *S. pombe* strain (close and distant relatives of *S. cerevisiae*, respectively). Both WT and RNase H2-defective backgrounds were used, as well as several additional *S. cerevisiae* strains (RED, a PIP-defective mutant RNase H2 mutant, and an *rnh1* mutant). This was a Herculean effort that generated a data-rich manuscript, and also may be the first example of mapping nuclear rNMPs in a WT background. rNMPs were examined in mtDNA, and data confirm the lack of RER in mitochondria and demonstrate a strong over-representation of rC in *S. cerevisiae* and *S. paradoxus* but rG in *S. pombe* (Figure 1). An additional difference is a bias for rNMP insertion in CG-rich regions in *S. cerevisiae* and *S. paradoxus*, but not in *S. pombe* (Figure 2). Finally, there is a clear effect of the previously incorporated dNMP on rNMP incorporation/identity (Figure 3). In terms of rNMP patterns/levels in nuclear DNA, the most striking results are the difference between WT and *rnh201* mutants (Figure 4) and the patterns of incorporation in repetitive DNA (Figure 5B). There are multiple important insights provided by the analyses, and only minor revisions are suggested.

1. There needs to be some comment in the main text concerning the choice of *S. cerevisiae* strains used.

2. If the authors are going to present formulas in Figures 1 and 4, they should provide some explanation in the main text. The normalized values will obviously be affected by concentrations in the corresponding rNTP pool. How will this affect the over- or under-representation of rNMPs?

3. A bit more discussion of possible RNase H2 preferences would be helpful.

4. Perhaps the under-representation of U reflects the preference of Top1 to incise at T?

5. Any ideas why there is a repetitive pattern of rNMPs in repeated sequence?

Minor issues

“Differently” is awkwardly used throughout.

On the middle of p. 14, should the reference be to Figures 1B and 4B instead of 1C and 4C, which are formulas?

Middle of p. 15 – motifs, not motives

Major points of the response to reviewers and/or modifications included in the revised manuscript:

- i) We provide a validation of the ribose-seq technique and the major findings presented in our manuscript by analyzing the data of Reijns *et al.*, *Nature* 2015 that have been generated utilizing the emRiboSeq technique for mapping ribonucleotides (rNMPs) in DNA. The emRiboSeq approach is completely distinct from ribose-seq in the preparation of the libraries with rNMP sites. Our analyses of the emRiboSeq data show results that are consistent with the major findings presented in our manuscript using ribose-seq. We have added these results in **Supplementary Figures 4 and 8** of the revised text. Such a comparison of the two datasets demonstrates that the major findings of our study are independent from the technique utilized.
- ii) We highlight that we analyzed repeats for many strains in multiple instances throughout our study. In fact, all our major conclusions are based on many consistently reproducible results.
- iii) We emphasize that there is not a single data point presented in our study that has been published before. We did reproduce a few of our previously published ribose-seq libraries as well as those from other groups, but only to use them as references to which to compare the results of our new libraries. All these comparisons further corroborate the validity of our technique and results. As stated by Reviewer 3, we conducted a “Herculean” work to significantly advance the knowledge about the biology of rNMPs in DNA. We feel that in some instances of the review our results have been misinterpreted, and thus the values and novelty of our findings have not been fully grasped.
- iv) Suggested analysis of dNTP and rNTP pools were conducted. The data obtained are supportive of specific rNMP patterns observed in our study. The results of these analyses are discussed in the manuscript and shown in **Supplementary Figure 1**. The measures of dNTP and rNTP pools were conducted by colleagues at Emory University (Drs. S. Tao, N. Bowen, R. Schinazi and B. Kim), who have been included among the authors of the manuscript.
- v) We believe that we clarified all the points of misinterpretation and rigorously addressed all the Referees’ criticisms and suggestions. Moreover, we hope that all the Reviewers will recognize the technical robustness of our approach for rNMP mapping, the richness of novel findings that we obtained, and the value that our study adds to the field of genome stability.

Thank you very much for your consideration.

Point-by-point-response plan to each comment of the reviewers:

Reviewer #1 (Remarks to the Author):

1) In this manuscript, the authors have used a single method, Ribose-seq, on DNA from *S. cerevisiae* *S. paradoxus* and *S. pombe*, to identify the base identity of incorporated ribonucleotides, the nucleotide-sequence context of incorporated ribonucleotides and highlight patterns of ribonucleotide in repeats in each strain. They also investigated if these parameters changed upon deletion of *rnh1*, *rnh201* or in strains with a *pip* mutant or RED-mutant in *S. cerevisiae*.

- The ribose-seq technique, coupled with the Ribose-Map bioinformatics toolkit have been recently developed by our team (Koh *et al.*, *Nature Methods* 2015, and Gombolay *et al.*, *Nucleic Acids Research* 2019, respectively). In this study, we have presented results obtained using the ribose-seq technique followed by Ribose-Map analysis. Because the Ribose-Map toolkit can easily process data from all other ribonucleotide (rNMP) mapping techniques, we have also conducted a data analysis utilizing the emRiboSeq technique (developed by A. Jackson group, Reijns *et al.*, *Nature* 2015, and detailed in Ding *et al.*, *Nat. Protocols* 2015). Among the rNMP mapping approaches, the one that is most diverse from our ribose-seq is emRiboSeq, because it does not fragment DNA by restriction enzymes, but by sonication, does not use alkali, and does not use AtRNase H2, but utilizes human recombinant RNase H2 and T4 quick ligase to cleave at rNMPs and capture the broken DNA ends upstream from the rNMPs, respectively. As described below, the results of emRiboSeq data analyses are much in line with those we obtained using ribose-seq that are presented in our manuscript. Comparison of data obtained analyzing these two very different methods (ribose-seq and emRiboSeq) demonstrates that the major findings of our study are independent from the technique utilized (see response to point 2 below and to Reviewer 2 from the beginning).

2) Several studies have already highlighted the base identity of incorporated ribonucleotides in yeast (Wanrooij *et al.*, Clausen *et al.*) and in humans (Berglund *et al.*, CF Moss *et al.*), however it has not been published that dA is preferred upstream of an incorporated ribonucleotides. In total they analyzed 34 different strains, but unfortunately there is no replicate of individual strains, and it is therefore not easy to judge if the observed variation is due to different strains or due to technical variation.

- Before the studies mentioned by this Reviewer, the studies by the A. Jackson, T. Kunkel and A. Carr groups, together with our group have all provided initial information about base identity of incorporated rNMPs in yeast DNA in 2015 (see News and Views by Jinks-Robertson and Klein, *NSMB* 2015 and the 4 specific articles cited in this publication). These data were preliminary and limited to 2 yeast backgrounds for *S. cerevisiae* and 1 for *S. pombe* (*S. cerevisiae* strain backgrounds used were $\Delta l(-2)l-7BYUNI300$ both in Clausen *et al.*, 2015 and in Reijns *et al.*, 2015, and E134 in Koh *et al.*, 2015; and *S. pombe* strain background was IM642 in Daigaku *et al.*, 2015). Wanrooij *et al.*, 2017 used mutant strains affecting concentrations of dNTP pools, all derived from *S. cerevisiae* strain background W1588-4c (derivative of W303), revealed an inverse correlation between concentration of dNTPs and their abundance in mitochondrial DNA and nuclear DNA of *rnh201*-null cells, and provided supportive evidence for lack of RNase H2 activity on rNMPs embedded in *S. cerevisiae* mitochondrial DNA. Despite these studies, there is still limited information regarding sites of rNMP incorporation in yeast genomic DNA, as well as other genomes, and how such sites are conserved among different strains and species. Berglund *et al.*, *PLoS Genetics* 2017 worked on human mitochondrial DNA of cell lines, supported role of DNA polymerase gamma in rNMP incorporation in human mtDNA, reported lack of activity of RNase H2 on human mtDNA, and also how nt pool imbalances can change the abundance of rA, rC, rG or rU in the human mtDNA. Moss *et al.*, *NAR* 2017 studied mitochondrial DNA with a focus on liver cells in the murine MPV17 disease model, reported strong dependence on rNTP/dNTP ratio on rNMP incorporation in mtDNA of solid tissues and increase in rGMPs following Mpv17 deficiency. Considering this last study and those of Hiller *et al.*, *Cancer Res.* 2018 and Aden *et al.*, *Gastroent.* 2019, which all link rNMP incorporation to pathology in mouse models, it is certainly of high value to learn more about the preferred sites of incorporation of rNMPs in DNA from yeast to mammalian cells.

Unexpectedly, we discovered that the nucleotides upstream from the rNMP, and, in particular, the nucleotide immediately upstream from the rNMP has major impact on rNMP distribution in yeast

genomic DNA. This finding does not derive from a simple experiment showing that “dA is preferred upstream of an incorporated ribonucleotide”. In, fact, as stated by Reviewer #3, the findings of our manuscript derive from a “Herculean” work employing: 3 different yeast species, 8 different yeast strain backgrounds (6 for *S. cerevisiae*, 1 for *S. paradoxus* and 1 for *S. pombe*), 9 different genotypes (5 for *S. cerevisiae*, 2 for *S. paradoxus* and 2 for *S. pombe*), with RNase H wild-type genotype repeated 11 times among the 6 strain backgrounds for *S. cerevisiae*, 3 times for *S. paradoxus* and 3 times for *S. pombe*, *rnh201*-null genotype repeated 8 times among 3 genotypes for *S. cerevisiae*, 1 time for *S. paradoxus* and 2 times for *S. pombe*, *rnh202-pip* genotype repeated 2 times using 2 *S. cerevisiae* strains, *rnh1*-null and *rnh201-RED* genotypes repeated each 2 times for 1 *S. cerevisiae* strain, and repeats of libraries from the same strain using different sets of restriction enzymes (2 for E134 -WT, 3 for BY4742-WT, 3 for YFP17-WT, 2 for E134-*rnh1*, 6 for E134-*rnh201*, 2 for BY4742-RED, 3 for DG2204-WT, 3 for JK105-WT and 2 for JK105-*rnh201*) for yeast mitochondrial data; and 3 different yeast species, 6 different yeast strain backgrounds (4 for *S. cerevisiae*, 1 for *S. paradoxus* and 1 for *S. pombe*), 9 different genotypes (5 for *S. cerevisiae*, 2 for *S. paradoxus* and 2 for *S. pombe*), with RNase H wild-type genotype repeated 6 times among 4 strain backgrounds for *S. cerevisiae*, 2 times for *S. paradoxus* and 2 times for *S. pombe*, *rnh201*-null genotype repeated 8 times among 3 genotypes for *S. cerevisiae*, 1 time for *S. paradoxus* and 2 times for *S. pombe*, *rnh202-pip* and *rnh1*-null genotypes repeated each 1 time using 1 *S. cerevisiae* strain, and *rnh201-RED* genotype repeated 2 times for 1 *S. cerevisiae* strain, and repeats of libraries from the same strain using different sets of restriction enzymes (2 for E134 -WT and 2 for BY4742-WT, 6 for E134-*rnh201*, 2 for BY4742-RED, 2 for DG2204-WT, 2 for JK105-WT, and 2 for JK105-*rnh201*) for yeast nuclear data. The comprehensive list of our yeast species, stains, genotypes, and ribose-seq mitochondrial and nuclear libraries with associated raw rNMP data is presented in Supplementary Tables 3 and 6, respectively.

Writing that “there is no replicate of individual strains” is incorrect. For example, Fig. 1b shows heatmaps of all our libraries (34) and the specific repeats of the same strain are grouped under the heatmap by curly brackets, please see Figures 1b, 3a,b and 4b and their legends, as well as Suppl. Figures 3A-I and 6A-I and their legends: <<*S. cerevisiae* libraries derived from the same strains are grouped together by curly brackets>>. For *S. paradoxus* and *S. pombe* we have repeats as shown in all these figures, as well as in Suppl. Tables 3 and 6. While Figures 1d-i, 2a-d and 4d-i show examples of plots, for which we added strain name, genotype and restriction enzyme set (specific libraries with strain name and different restriction enzyme set used are indicted in the legends), plots obtained from all the libraries are shown in Supplementary Figures 1A-H and 5A-H. Supplementary Tables 3,5-7 also show data obtained with all libraries. It is a major emphasis of our study to highlight data that are reproducible and occurring multiple times across different repeats of the same strain and genotype, across different strains of the same species, and across different species of yeast. The fact that the nucleotide upstream from the rNMP is the one showing strongest impact on the rNMP distribution in DNA of yeast cells is a conclusion that we took by analyses of all our libraries in the i) nucleotide plot data, ii) heatmap data, iii) MEME program data and iv) genome browser data.

We modified the text of Introduction of the manuscript to better highlight the value and reproducibility of our results: page 2, line 14 << The first studies reporting rNMP identity and distribution at the genome level used two yeast backgrounds for budding yeast *Saccharomyces cerevisiae* and one for fission yeast *Schizosaccharomyces pombe*, in which the gene coding for the catalytic subunit of RNase H2 (*RNH201*) was inactivated.>>, on page 3, line 1: <<Interestingly, recent studies directly linked rNMP incorporation to pathology in mouse models (Aden et al., 2019; Hiller et al., 2018; Moss et al., 2017). Therefore, it is of high value to learn more about the preferred sites of incorporation of rNMPs in DNA from yeast to mammalian cells. >>, and page 3, line 15 << By analyzing data from 34 ribose-seq libraries, which

include also a comparative examination with five emRiboSeq libraries prepared in (Reijns et al., 2015), we uncovered reproducible new features and patterns, ...>>. As well as that of the Discussion on page 14, line 13: << Our results, observed consistently across multiple genotypes and/or strains, show that rNMPs are found at preferential sites in DNA, rather than being randomly distributed along the mitochondrial and nuclear genomes. >>, and page 17, line 15: << In conclusion, via mapping and genome-wide analysis of a large number of ribose-seq libraries with rNMPs across three yeast species with wild-type and RNase H2 mutant genotypes, which generated consistently reproducible results, we revealed several new biological features, and a fundamental rule shaping the patterns of rNMP incorporation in yeast DNA. >>.

Moreover, we reproduced our major findings that the nucleotide upstream of the rNMP is the one showing strongest impact on the rNMP distribution in DNA of yeast cells by analyzing data generated using a technique that is completely different from ribose-seq: emRiboSeq. The emRiboSeq technique (by A. Jackson group, Reijns *et al.*, *Nature* 2015, detailed in Ding *et al.*, *Nat. Protocols* 2015) does not use restriction enzymes, alkali and AtRNL like ribose-seq, but sonication, recombinant human RNase H2 and T4 quick ligase to fragment the DNA, cleave at rNMPs and capture the nucleotide upstream from the rNMP, respectively. We collected published data of Reijns *et al.*, *Nature* 2015, and analyzed five emRiboSeq libraries (SRR1734967, SRR1734969, SRR1734972, SRR1734980 and SRR1734982 that all have wild-type DNA polymerase genes) derived from *rnh201*-null cells of background strain $\Delta(-2)l-7BYUNI300$. No wild-type RNase H2 cells were used in Reijns *et al.*, *Nature* 2015. Analyses of the *S. cerevisiae* emRiboSeq data for mitochondrial and nuclear DNA (see Methods) provide further support to our major conclusions. These new results are shown in the revised manuscript in **Suppl. Figures 4A-J and 8A-C**. For the mononucleotides, background $\Delta(-2)l-7BYUNI300$ of the emRiboSeq libraries has more similarity to the BY4742 strain that we used in ribose-seq libraries, with rC and rG prevalent and rU as the lowest after normalization in mitochondrial DNA, and also higher rC and rG and lowest rU for nuclear DNA (**Suppl. Figure 4A,B**). Striking correspondence is seen for the dinucleotide heatmaps, particularly for the mitochondrial data. Compare **Suppl. Figure 4C and 4D** with Fig. 3a and 3b of our manuscript, respectively, as well as **Supplementary Figures 4E and 4F** with Suppl. Fig. 3A and Suppl. Fig. 6A, respectively, **Supplementary Figure 4G,H** with Suppl. Fig. 3B,C, and **Supplementary Figure 4I,J** with Suppl. Fig. 6b,c. These new data show that the nucleotide immediately upstream from the rNMPs has most impact on rNMP distribution also in the emRiboSeq data. Importantly, this pattern is conserved between emRiboSeq and ribose-seq mitochondrial and nuclear libraries for *S. cerevisiae rnh201*-null cells. We also built MEME plots for the top 1% reads of mitochondrial emRiboSeq libraries (same as we did for our ribose-seq libraries described in Fig. 5a,b) showing hotspot motif with the rNMP and the nucleotide immediately upstream of the rNMP (compare **Suppl. Figure 8A,B** with Fig. 5a and Suppl. Fig. 7A, respectively). Moreover, we found a hotspot site within the repeat region on locus ChrM: 63,563-63,672 for emRiboSeq library SRR1734980, which has rG hotspot within the TAAGTA-repeated sequence on the forward strand, and in part the rC sticks out on the complementary strand (**Suppl. Figure 8C**), similarly to what we show in Fig. 5d. The same hotspot in the same repeated sequence was found also in emRiboSeq library SRR1734982. All these new results obtained analyzing the emRiboSeq libraries are presented in the Results and Discussion.

The fact that we noted some diversity among the *S. cerevisiae* strains that we analyzed is also supported by multiple repeats of results. Among the 6 strains of *S. cerevisiae* that we analyzed (E134, BY4741, BY4742, YFP17, W303 and S288C), E134 was the one with less similarity to all other 5 strains at the mitochondrial level. For the mitochondrial libraries of the E134 strain we have 2 wild-type, 1 *rnh202-pip*, 2 *rnh1*-null, and 6 *rnh201*-null libraries, a total of 11 E134 libraries, which all are very similar to each other and show less rG than the libraries derived from the other *S. cerevisiae* strains in the mononucleotide heatmaps (see Fig. 1b). As suggested by Reviewers #2 and 3, we examined whether this

difference in rNMP composition for E134 relative to the other *S. cerevisiae* strains was linked to a particular composition of nucleotide pools for E134. We analyzed and compared the composition of dNTPs and rNTPs obtained from E134 strain with that obtained from BY4742. These results are shown in **Suppl. Figure 1** of the revised manuscript, and are discussed on page 4, line 18 << By measuring rNTPs/dNTPs ratios for strains E134 and BY4742, we found that the rGTP/dGTP ratio was significantly lower in E134 compared to BY4742 ($P = 0.0008$) (**Supplementary Figure 1**), possibly accounting for the lower rG incorporation in E134. >> and in the Discussion on page 15, line 23 << Lower ratio for rGTP/dGTP in E134 compared to BY4742 strain may account for such preference>>.

Other comparisons of rNMP patterns across libraries are: rNMPs in C/G-rich tracts of budding yeast libraries and not in fission yeast libraries (see example in Fig. 2 together with Suppl. Fig. 1A-H), low rU across mitochondrial DNA in libraries of all genotypes, backgrounds, and species, low rU in all nuclear *rnh201*-null libraries used, low rG in all wild-type nuclear DNA libraries of all the 3 yeast species including the *rnh202-pip* and *rnh1*-null mutants of *S. cerevisiae*, and dA most often preceding rG both in *S. cerevisiae* and *S. paradoxus* mitochondrial libraries, and in all nuclear libraries across all the three yeast species for all strains and genotypes examined in this study. All these findings are based on large numbers of reproducible results. In sum, we have major conclusions that are derived from multiple libraries and repeats of experiments.

3) Figure 1a and b and Figure 4a and b have been published before for *S. cerevisiae*.

- This statement is incorrect. There is not a single data point presented in these figures and in our study in general that has been published before. All the material presented in this study is original and new. Figures 1a,b and 4a,b show comparisons of rNMP distribution and patterns among the three yeast species of *S. cerevisiae*, *S. paradoxus* and *S. pombe* in wild-type and *rnh201*-null cells. To our knowledge, such comparisons have never been made before. For the purpose of making valuable comparisons among the rNMP patterns of the three different yeast species of *S. cerevisiae*, *S. paradoxus* and *S. pombe* in the wild-type and *rnh201*-null genotypes, we built de-novo libraries from all these yeasts, and obtained new sets of data also for *S. cerevisiae* *rnh201*-null cells, which serve as reference and corroborate some of the previous data obtained for this genotype of *S. cerevisiae*.

4) The manuscript does not provide much further biological function of ribonucleotides in DNA and I do not believe this study add enough news value to the already published work.

- To our knowledge, this study is the first to provide distinct evidence that the DNA sequence context shapes the pattern and distribution of rNMPs in genomic DNA. rNMPs in DNA constitute the most common alteration of DNA structure; their intrusion in DNA changes DNA properties and is often associated with mutations and genomic instability. Although rNMPs are abundantly present in DNA, it remained largely unknown whether rNMPs of a specific base are randomly incorporated in DNA, whether there are preferred sites of rNMP incorporation in the genome, and whether rNMP sites and/or patterns are conserved among different species or cell types from the same species. In our study, we uncovered a fundamental rule shaping the patterns of rNMP incorporation in yeast DNA. Genomic rNMP incorporation not only is far from being random, but also is markedly driven by the sequence context; specifically, by the sequence of the nucleotide immediately upstream from the site of incorporation. Knowing that there are preferred sites of rNMP incorporation in DNA may have important implications to explore the sequence context of rNMP incorporation in other cell types, with a focus on human cells, like cells derived from Aicardi Goutières patients or cancer cells to identify sites of fragility in DNA. In

addition, we find that short nucleotide repeated tracts of specific sequences can be hotspot sites of rNMP incorporation. Furthermore, we revealed both conservation and variation among the rNMP patterns of different yeast species and strains. The budding yeasts have strong similarities in rNMP patterns with one another, while the fission yeast is more like the budding yeasts at the nuclear than at the mitochondrial level. Overall, we uncovered several new features, as well as conserved motifs, patterns and hotspots of rNMP in yeast DNA, which strikingly all highlight a major role played by the upstream dNMP in rNMP incorporation. The presented results provide important novel mechanistic understanding of the process of rNMP incorporation in DNA and open a new perspective in the molecular relationship between ribonucleotides and deoxyribonucleotides in cells. Therefore, we believe that our study is of significant value and interest to the scientific community and the broad audience of *Nature Communications*. As indicated in point 2 above, to emphasize the value of our study, we edited the text of Introduction and Discussion as follows on page 2, line 14 << The first studies reporting rNMP identity and distribution at the genome level used two yeast backgrounds for budding yeast *Saccharomyces cerevisiae* and one for fission yeast *Schizosaccharomyces pombe*, in which the gene coding for the catalytic subunit of RNase H2 (*RNH201*) was inactivated.>>, on page 3, line 1: <<Interestingly, recent studies directly linked rNMP incorporation to pathology in mouse models (Aden et al., 2019; Hiller et al., 2018; Moss et al., 2017). Therefore, it is of high value to learn more about the preferred sites of incorporation of rNMPs in DNA from yeast to mammalian cells.>>, on page 3, line 15 << By analyzing data from 34 ribose-seq libraries, which include also a comparative examination with five emRiboSeq libraries prepared in (Reijns et al., 2015), we uncovered reproducible new features and patterns, ...>>, on page 14, line 13: << Our results, observed consistently across multiple genotypes and/or strains, show that rNMPs are found at preferential sites in DNA, rather than being randomly distributed along the mitochondrial and nuclear genomes.>>, and page 17, line 15: << In conclusion, via mapping and genome-wide analysis of a large number of ribose-seq libraries with rNMPs across three yeast species with wild-type and RNase H2 mutant genotypes, which generated consistently reproducible results, we revealed several new biological features, and a fundamental rule shaping the patterns of rNMP incorporation in yeast DNA.>>.

Minor comments:

Abstract, There are certainly significant conservation of ribonucleotide incorporation, but how often are specific sites imprinted with ribonucleotides?

- It is the scope of this study to characterize sites of ribonucleotide incorporation and their surrounding sequence context rather than quantify ribonucleotides in DNA. The frequency of rNMP incorporation varies depending on the site, with certain sites having markedly increased incorporation compared to others. Since there are sites of markedly increased incorporation of rNMPs, we chose to study potential biological signatures of these sites. To study such sites, we identified the top 1% of sites that have frequent rNMP incorporation in each ribose-seq library. Based on the top 1% of sites, we were able to identify consensus sequences that are common to each site, and also found that these consensus sequences are conserved among our replicates (Fig. 5a,b of our manuscript). In addition to studying sites where rNMP incorporation is markedly abundant, we also studied sites of rNMP incorporation that are common to all of our libraries (Suppl Table 7A-C). From this analysis, we found hotspots in both the nuclear and mitochondrial DNA that were present in all of our libraries. Such reproducibility suggests that these sites might be biologically significant sites of rNMP incorporation. To better clarify this point we edited the text on page 12, from line 17: << To determine whether there were overlapping features and specific signatures among rNMPs that were most frequently incorporated in mitochondrial and nuclear DNA, we

selected the top 1% most abundant rNMP sites from each mitochondrial and nuclear library and analyzed them using the MEME program. Since the number of rNMPs at a particular site can vary depending on sequencing coverage, calculating the top 1% allowed us to compare the frequency of rNMPs at each site among each of our libraries independently of the sequencing coverage of those particular libraries. >>.

Page 3, What are the substantial modifications to the original Ribose-seq and how does that change the data?

- We have optimized and improved our ribose-seq protocol. Most of details are described in the book chapter by Balachander & Yang *et al.*, *Meth Mol. Biol.* that is now published, which is reference #29 in our manuscript, and which we initially provided with our submitted manuscript. We edited the Supplementary Methods to highlight the most relevant modifications that were implemented in this study. We added this text in the Suppl. Methods on page 31, from line 9: <<...we optimized the ribose-seq protocol by i) redesigning the molecular barcode-containing adaptor, making it shorter and removing overlapping sequences, ii) fragmenting the genome of interest in smaller fragments (~450 bp); iii) performing two rounds of PCR and overall reducing the PCR cycle number; iv) cutting and purifying a specific size-range of the ribose-seq library from the nondenaturing gel to eliminate any primer dimers formed during PCR and any long products that are not proficient for sequencing>> .

Page 3, Supplementary table 3-5 is missing.

- We did upload all Supplementary Tables, including Suppl. Tables 3-5 in our submission. Because these are Excel files, it is possible that, beyond our control, there has been some issue after the uploading or downloading of these tables. We have re-uploaded all Supplementary Tables.

Page 11, The authors analyze so called hotspots of ribonucleotide incorporation, these hot spots could be generated during PCR amplification so it would be important to show that these are unique hotspots and not only PCR products which are amplified more efficient.

- As described in the Supplementary Methods, in the section on “Processing and alignment of sequencing reads” << the sequencing reads consist of an eight-nucleotide UMI, a three-nucleotide molecular barcode, the tagged nucleotide (the nucleotide tagged during ribose seq from which the position of the rNMP is determined), and the sequence directly downstream from the tagged nucleotide. The UMI corresponds to sequencing cycles 1-6 and 10-11, the molecular barcode corresponds to cycles 7-9, the tagged nucleotide corresponds to cycle 12, and the tagged nucleotide’s downstream sequence corresponds to cycles 13+ of the raw FASTQ sequences. ...The Alignment Module de-multiplexed the trimmed reads by the appropriate molecular barcode, aligned the reads to the reference genome using Bowtie 2, and de-duplicated the aligned reads using UMI-tools>>. Thus, all reads that are analyzed are not PCR duplicate because they have different UMI. For the hotspot tracts in the short-nucleotide repeated tracts, in the text of the manuscript, in the legend of Figure 5d on page 23 line 6 we write: <<All reads shown here have distinct UMI, meaning they do not represent PCR duplicates caused by slippage of DNA polymerase>>, and similarly, in the Results page 13, line 22 we highlighted that: << These are not artifacts of PCR because all reads in a given short-nucleotide repeated tract region have different UMI, not just those reads that are at the same position within the repeated tract>>.

Page 12, How did you analyze the libraries in the genome browser?

- As discussed in the Supplementary methods in the ‘Genome Browser’ section, BedGraph files were generated using the Distribution Module of Ribose-Map and then visualized using the JBrowse genome browser. Beyond applying the MEME program and custom scripts that analyze nucleotide sequence context to the genome browser data, we also browsed the genomic data of ribose-seq libraries chromosome by chromosome to identify sites of interest. The concentration of rNMPs at defined positions within specific short-nucleotide repeated tracts was a feature that emerged. To clarify this point, we edited the text of the Results at the beginning of the chapter ‘Patterns of rNMPs in short nucleotide repeats reveal ArC and ArG as common hotspots’: <<Concentration of rNMPs at short-nucleotide repeated tracts was a feature that emerged by browsing the genomic data of the ribose-seq libraries...>>. Moreover, for the analysis of short nucleotide repeat tracts, we only considered reads that were longer than the repeated regions to ensure accuracy of our findings, see text on page 38, line 12: << For the analysis of short nucleotide repeat tracts, we only considered reads that were longer than the repeated regions to ensure accuracy of our findings>>.

Page 3, Supplementary: The ribose-seq protocol lacks volumes and units used for each enzyme, nucleotide concentrations, adapter concentrations and the PCR program used.

- We did not want to overload the manuscript with details that are already present in our publications, like in Balachander & Yang *et al.*, in *Meth Mol Biol* 2019 (Ref #29). As suggested by this Reviewer, in the section ‘Ribose-seq library preparation’, we added these details: << The single-stranded (ss) DNA fragments were incubated with 1 μ M *Arabidopsis thaliana* tRNA ligase (AtRNL), 50 mM Tris-HCl pH 7.5, 40 mM NaCl, 5 mM MgCl₂, 1 mM DTT and 300 μ M ATP in a volume of 20 μ L for 1 h at 30 °C, followed by purification>>, << The fragments were then treated with T5 Exonuclease (NEB) 50 units in 50 μ L volume for 1 h 30 min at 37 °C to degrade the unligated ssDNA fragments>>, << the circular fragments were incubated with 1 μ M 2’-phosphotransferase (Tpt1), 20 mM Tris-HCl pH 7.5, 5 mM MgCl₂, 0.1 mM DTT, 0.4% Triton X-100 and 10 mM NAD⁺ in a volume of 40 μ L for 1 h at 30 °C>>, << the circular fragments were PCR-amplified using two rounds of amplifications to result in ribose-seq library: both PCR rounds begin with an initial denaturation at 98 °C for 30 sec. Then denaturation at 98 °C for 10 sec, primer annealing at 65 °C for 30 sec, and DNA extension at 72 °C for 30 sec are performed. These 3 steps are repeated for 6 ~ 15 cycles in the first PCR round, and for 7 ~ 13 cycles in the second PCR round depending on the concentration of the circular ssDNAs containing the rNMPs. Successively, there is a final extension reaction at 72 °C for 2 min for both PCRs>>. We further emphasized that all specific details and tips for the ribose-seq protocol are available in *Meth Mol Biol* 2019 (Ref #29).

Figure 1d-i, labels with each strain is missing.

- This information was not missing; it was provided in the legend with the genotype and library name. Now, we also added the strain name, genotype and restriction enzyme set in the Figure 1d-i, as well as in its legend: <<Zoomed-in plots of normalized nucleotide frequencies relative to mapped positions of sequences from mitochondrial ribose-seq libraries. Plots derived from example libraries are shown for *S. cerevisiae* (d) WT (YFP17-WT-RE2-FS146) and (e) *rnh201* Δ (BY4742-*rnh201*-RE2-FS138), *S. paradoxus* (f) WT (DG2204-WT-RE1-FS108) and (g) *rnh201* Δ (DG2204-*rnh201*-RE1-FS130), and *S. pombe* (h) WT (JK105-WT-RE1-FS94) and (i) *rnh201* Δ (JK105-*rnh201*-RE3-FS135)>>. The same applies for Figure 4d-i and its legend.

Figure 2: labels with each strain is missing.

- This information was not missing; it was provided in the legend with the genotype and library name. Now, we also added the strain name, genotype and restriction enzyme set in the Figure 2, as well as in its legend: << Shown are examples of combined and single rNMP plots of normalized nucleotide frequencies relative to mapped positions of rNMPs from examples of mitochondrial ribose-seq libraries. (a) Zoom-out and (b) zoom-in plots for mitochondrial library BY4742-WT-RE1-FS104 of *S. cerevisiae* WT cells. (c) Zoom out plots for mitochondrial library DG2204-WT-RE1-FS108 of *S. paradoxus* WT cells. (d) Zoom-out plots for mitochondrial library JK105-WT-RE1-FS94 of *S. pombe* WT cells>>.

Figure 3: why are there fewer strains used in panel b, compared to a?

- This information was provided in the Supplementary Methods and is on page 35, line 25: <<All mitochondrial libraries (34/34) had very low background (0.04% - 4.85%). Majority of the nuclear libraries (25/34) had background <12% (0.02% -11.74%), and these were studied>>.

Reviewer #2 (Remarks to the Author):

In this study, Balachander et al. performed systematic analysis of ribonucleotide identity and their sequence context in the genome using Ribose-seq approach previously developed by the Storici group. Their analysis includes two *Saccharomyces* species and the *Schizosaccharomyces pombe*. Consistent with previous reports, they confirm preferential incorporation/retention of certain bases over others, eg. preference for rC in RNase H2-deficient *Saccharomyces*. They uncovered that the ribonucleotide incorporation is influenced by immediate sequence upstream of rNMP, eg. dA is preferred upstream of rC and rG. These are interesting observations. However, it is not immediately clear what the implication may be for the biology of ribonucleotide incorporation. Moreover, in my opinion, validation of the overall approach has yet to be established.

- We thank the Reviewer for the thoughtful comments. As discussed above, we performed a validation of our major results by employing the Ribose-Map bioinformatics toolkit to analyze the published data of Reijns *et al.*, *Nature* 2015, which made use of an rNMP mapping approach (emRiboSeq) that is completely different from ribose-seq. The emRiboSeq approach fragments DNA by sonication and not restriction enzymes and utilizes human recombinant RNase H2 to cleave at rNMP sites, while ribose-seq utilizes alkali. Moreover, T4 quick ligase is used in emRiboSeq to ligate the nucleotide upstream of the rNMP to the adaptor sequence, while the rNMP terminated DNA ends are directly ligated to the adaptor sequence by AtRNL in the ribose-seq approach. We analyzed all 5 emRiboSeq libraries (SRR1734967, SRR1734969, SRR1734972, SRR1734980 and SRR1734982) prepared by Reijns *et al.* from *S. cerevisiae rhh201*-null cells of background strain $\Delta l(-2)l-7BYUNI300$ with wild-type DNA polymerase alleles. We prepared heatmaps for mono and dinucleotides for these 4 libraries. The results are shown in **Supplementary Figure 4A-J** in the revised manuscript. The mitochondrial results from the emRiboSeq libraries are almost identical to those we obtained with our ribose-seq libraries. While the nuclear data obtained using the emRiboSeq approach have some variation with those obtained using ribose-seq, still there is good similarity between the two, especially with our BY4742 strain, and again the

deoxyribonucleotide upstream from the rNMP is the one having most impact on rNMP incorporation also for the emRiboSeq data. We also built MEME plots for the top 1% reads of mitochondrial emRiboSeq libraries showing hotspot motif with the rNMP and the nucleotide immediately upstream of the rNMP (compare **Supplementary Figure 8A,B** with Fig. 5a and Supplementary Fig. 7a). Moreover, we found a hotspot site within the repeat region on locus ChrM: 63,563-63,672 for emRiboSeq library SRR1734980, which has rG hotspot within the TAAGTA-repeated sequence on the forward strand, and in part the rC sticks out on the complementary strand (**Supplementary Figure 8C**), similarly to what we show in our Fig. 5d. These results obtained with the analyses of the emRiboSeq libraries are discussed in the Introduction on page 3, line 15: << By analyzing data from 34 ribose-seq libraries, which include also a comparative examination with five emRiboSeq libraries prepared in (Reijns et al., 2015), we uncovered reproducible new features and patterns,...>>, in the Results on page 7, from line 1: << Moreover, to ensure that results obtained for the dinucleotides were not an artefact of the ribose-seq technique, we exploited Ribose-Map (Gombolay et al., 2019) to analyze data from four libraries of *S. cerevisiae* *rnh201*-null cells prepared using the emRiboSeq technique, which does not employ restriction enzymes, alkali and AtRNL, but sonication, human recombinant RNase H2 and T4 quick ligase, respectively (Reijns et al., 2015). We generated heatmaps for the mononucleotides and examined the dNMPs at positions -1, +1, -2 and +2 relative to the rNMPs in these libraries (Supplementary Fig. 4A,C,E,G,H). The similarity of the emRiboSeq results to those obtained using ribose-seq is remarkable. This analysis further supports our findings that the dNMP upstream of the rNMP has the most impact on the incorporation of a specific rNMP type in yeast mitochondrial DNA.>>, page 10, line 19: << Analysis of the four *rnh201*-null libraries prepared using emRiboSeq revealed good correspondence with the biases observed in our ribose-seq libraries of the same genotype. The strongest biases were seen for the -1 dNMP (Supplementary Fig. 4B,D,F,I,J). >>, page 13, line 7: << Strikingly, deoxyribonucleotide with base A was dominant at the -1 position in all hotspot motifs for all mitochondrial libraries of *S. cerevisiae*, *S. paradoxus* and *S. pombe*, as well as in the *S. cerevisiae* emRiboSeq libraries (Supplementary Fig. 8A,B). >>, on page 13, line 10: << Although less pronounced, a dTMP was conserved at position -3 and +2 in most of the mitochondrial *S. cerevisiae* libraries, including the emRiboSeq libraries, >>, and on page 14, line 6: << Interestingly, this pattern was also evident in emRiboSeq libraries SRR1734980 (Supplementary Fig. 8C) and SRR1734982>>.

To our knowledge, there are no studies before ours providing marked evidence that the DNA sequence context shapes the pattern and distribution of rNMPs in genomic DNA. rNMPs in DNA constitute the most common alteration of DNA structure, their intrusion in DNA changes DNA properties and is often associated with mutations and genomic instability. Every DNA polymerase that has been examined for its capacity to incorporate rNMPs does so. Every genomic DNA that has been examined for the presence of rNMPs in DNA showed to contain rNMPs. The presence of rNMPs in genomic DNA appears to be a universal phenomenon in nature of which we currently understand only a minimal part. Despite rNMPs are abundantly present in DNA, it remained largely unknown whether rNMPs of a specific base are randomly incorporated in DNA, whether there are preferred sites of rNMP incorporation in the genome, whether rNMP sites and/or patterns are conserved among different species, or among different cell types from the same species. In our study, we uncovered a fundamental rule shaping the patterns of rNMP incorporation in yeast DNA. Genomic rNMP incorporation not only is far from being random, but also is markedly driven by the sequence context, specifically by the sequence of the nucleotide immediately upstream from the site of incorporation. Knowing that there are preferred sites of rNMP incorporation in DNA has profound implications to identify sites of genomic fragility, which may play a role in DNA mutability and/or evolution. As noted above in response of point 2 of Reviewer #1, recent studies have linked rNMP incorporation in DNA to pathology using mouse models (Moss *et al.*, *NAR* 2017, Hiller *et*

al., *Cancer Res.* 2018, and Aden *et al.*, *Gastroent.* 2019). Thus, we believe it is of high value to learn more about the preferred sites of incorporation of rNMPs in DNA from yeast to mammalian cells. For example, our finding that short-nucleotide repeated tracts are hotspot sites of rNMP incorporation suggests that rNMPs in these sites may increase the instability of these DNA regions. Furthermore, we revealed both conservation and variation among the rNMP patterns of different yeast species and strains. The budding yeasts have strong similarities in rNMP patterns with one another, while the fission yeast is more like the budding yeasts at the nuclear than at the mitochondrial level. Overall, we uncovered several new features, as well as conserved motifs, patterns and hotspots of rNMP in yeast DNA, which strikingly all highlight a major role played by the upstream dNMP in rNMP incorporation. The presented results provide important novel mechanistic understanding about the process of rNMP incorporation in DNA and open a new perspective in the molecular relationship between ribonucleotides and deoxyribonucleotides in cells. As discussed in response to Reviewer 1, we modified the text of Introduction and Discussion of the manuscript to better highlight the value and reproducibility of our results: page 2, line 14 << The initial studies reporting rNMP identity and distribution at the genome level used two yeast backgrounds for budding yeast *Saccharomyces cerevisiae* and one for fission yeast *Schizosaccharomyces pombe*, in which the gene coding for the catalytic subunit of RNase H2 (*RNH201*) was inactivated.>>, on page 3, line 1: <<Interestingly, recent studies directly linked rNMP incorporation to pathology in mouse models (Aden *et al.*, 2019; Hiller *et al.*, 2018; Moss *et al.*, 2017). Therefore, it is of high value to learn more about the preferred sites of incorporation of rNMPs in DNA from yeast to mammalian cells. >>, on page 3, line 15 << By analyzing data from 34 ribose-seq libraries, which include also a comparative examination with five emRiboSeq libraries prepared in (Reijns *et al.*, 2015), we uncovered reproducible new features and patterns, ...>>, on page 14, line 13: << Our results, observed consistently across multiple genotypes and/or strains, show that rNMPs are found at preferential sites in DNA, rather than being randomly distributed along the mitochondrial and nuclear genomes. >>, and page 17, line 15: << In conclusion, via mapping and genome-wide analysis of a large number of ribose-seq libraries with rNMPs across three yeast species with wild-type and RNase H2 mutant genotypes, which generated consistently reproducible results, we revealed several new biological features, and a fundamental rule shaping the patterns of rNMP incorporation in yeast DNA. >>.

Specific concerns and suggestions:

1. Enzymatic reactions involving nucleotide sequences have potential biases. Since the authors are examining arguably small to moderate differences in rNMP abundance and sequence context preferences, the intrinsic bias of the library preparation, if true, can significantly alter the results. The relative abundance of rNMP has been reported in some of the yeast species using alternative rNMP mapping techniques. This allows for comparison of observations using different approaches. A previous study using the PU-seq approach (Daigaku *et al.* 2015) reports that rA and rU are the most abundant rNMPs in RNase H2 deficient-*S. pombe* cells (Figure S3). In contrast, the current study identified rA as the most abundant and rU as the least abundant. The sequence context preference report is also somewhat different between the two studies. This apparent discrepancy is perhaps due to the difference in the intrinsic biases in different techniques. Ligation reactions often has been documented for several ligation-based Illumina library preparation methods. Thus, without further validation, it is unclear whether the sequence preferences, which is the central focus in this study, is due to bias in ribonucleotide incorporation or bias of the approach, or perhaps more likely, a combination of both.

- We thank the Reviewer for her/his comment and fully agree with the above statements. As noted above, we performed a validation of our major results analyzing available published data prepared using the emRiboSeq approach, which is the current rNMP mapping approach that is most different from ribose-seq, relying on human recombinant RNase H2 instead of alkali, and which does not make use of AtRNL but T4 quick ligase to capture the sequence upstream of the rNMP sites. The analyses of the emRiboSeq data revealed strong sequence preferences displaying biases of rNMP incorporation that are very similar to our ribose-seq data (**Supplementary Figures 4A-J** compared to Fig. 1b, 4b, 3a,b and Suppl. Fig. 3A, Suppl. Fig. 6A, Suppl. Fig. 3B,C and Suppl. Fig. 6B,C, respectively, as well as compare **Supplementary Fig. 8A-C** to Fig. 5a, Suppl. Fig. 7A and Fig. 5d, respectively).

rU was found to be the least abundant rNMP in our study in *rnh201*-null cells in all three species that we analyzed (*S. cerevisiae*, *S. paradoxus* and *S. pombe*). In addition, we searched for publicly available rNMP data of *Schizosaccharomyces pombe* and found this NCBI link from the T. Kunkel group: <https://www.ncbi.nlm.nih.gov/geo/query/acc.cgi?acc=GSE125855> from publication by Zhou *et al.*, *Nat. Comm.* 2019. We found 3 libraries of *rnh201*-null *S. pombe* in this link (libraries GSM3583288_TAK_1943, GSM3583292_TAK_1950 and GSM3583311_TAK_2034 that have wild-type DNA polymerase alleles). No wild-type RNase H2 data were present. We applied Ribose-Map and show here in this response bar graphs with normalized percentages of rNMPs for mitochondrial and nuclear data of these 3 libraries of *rnh201*-null *S. pombe* (**Figure R1a,b** below). These data have similarity with our results for *rnh201*-null *S. pombe*. rU is also low in the nuclear libraries of this Kunkel's study, in line with our results. The *S. pombe* strain background that we used is JZ105, the one used in the Daigaku *et al.*, *Nat. Struct. Mol. Biol.* 2015, as well as in Zhou *et al.* 2019, is W1588-4c. While we consistently found very low rU in nuclear DNA of *rnh201*-null cells of all yeast stains and species examined, it is possible that there is variation among strains, as we also revealed in our work for *S. cerevisiae* strains. It is possible that the difference in abundance of rU in nuclear DNA between the study of Daigaku *et al.*, and our work and the Kunkel's study is linked to the background of strains used. In addition, other factors that likely have impact on abundance of rU in DNA are some proofreading activity of DNA Pol delta on rU, activity of topoisomerase 1, as also pointed out by Reviewer #3, and variation in nucleotide pools.

Moreover, as this Reviewer and Reviewer #3 suggest, we performed analysis of dNTP and rNTP pools for the *S. pombe* strain and two *S. cerevisiae* strains. These data are shown in Supplementary Figure 1, and are discussed on page 8, line 2 << Like in *S. cerevisiae* cells, the rUTP/dTTP ratio was also found to be the lowest among rATP/dATP, rCTP/dCTP and rGTP/dGTP ratios in *S. pombe* cells of strain JK105 (Supplementary Fig. 1)>> and on page 16, line 2: << While the low level of rU can be explained by general high concentration of dTTP in the nucleotide pools (Clausen *et al.*, 2013; Wanrooij *et al.*, 2017) that we also observed in the *S. pombe* strain...>>.

2. In the Supplementary Methods, it was stated that "AtRNL cannot capture abasic sites or ... DNA breaks". However, in later sections, the authors acknowledged that reads from restriction sites contributed to background noise, apparently in some cases, up to or over 12% of the total reads. This contradicts the statement about the specificity of AtRNL. If the restriction sites contribute to the noise, then other random nicks and breaks can also be captured. One way to address the issue is perhaps to construct control libraries without alkali treatment.

- We thank the reviewer for this comment. We realized we did not clearly explain the point. The observed noise that we describe is unlikely due to activity of AtRNL on restriction enzyme cut ends but rather to

residual activity of T4 DNA ligase used to attach the adaptors in the initial steps of ribose-seq. The activity of AtRNL has been well studied in Schutz *et al.*, *RNA* 2010. Moreover, in Koh *et al.*, *Nat. Methods* 2015 we further examined the activity of AtRNL to ligate rNMP terminated DNA ends vs DNA ends with no ribose at the 3' end, and we found no activity in the absence of the 3'-terminal rNMP (see **Figure R2** below, which is Fig. 1a in Koh *et al.*, *Nat. Methods* 2015). We also always include a control sample w/o AtRNL for all the ribose-seq libraries that we prepare, and we see no smear on the PAGE gel for this control. These results have been reproduced many times in our ribose-seq preparations (see for example Koh *et al.*, *Nat. Methods* 2015 in Suppl Fig. 3A, shown here as **Figure R3a**, and Figs. 5 and 6 in Balachander & Yang *et al.*, *Meth. Mol. Biol.* in press, obtained using the optimized ribose-seq protocol, attached here below as **Figure R4a,b**). To better clarify this point, we modified the text of the Supplementary Methods on page 32, line 11 as follows: <<Due to the efficient removal of rNMPs by RNase H2, nuclear libraries of wild-type *RNH201*, *rnh202-pip* and *rnh1*-null cells generally had a much lower number of reads compared to the mitochondrial libraries of the same cells, and thus had a higher number of background reads that originated from the capture of restriction enzyme ends likely by residual activity of T4 DNA ligase>>.

Moreover, we do agree that it is important control to prepare ribose-seq libraries w/o alkali treatment. In fact, this is exactly what we first did in Koh *et al.*, *Nat. Methods* 2015. In Suppl. Fig. 3b of Koh *et al* 2015, we clearly showed that w/o alkali treatment we do not obtain a ribose-seq library. We present this same Figure here below as **Figure R3b**. In the revised manuscript, we also provide an additional example obtained using the optimized ribose-seq protocol, showing that no smear is obtained w/o alkali treatment (**Supplementary Fig. 10**). This further demonstrates that AtRNL does not work on DNA ends that lack a 3'-terminal rNMP. For better clarification, we modified the text on page 33, line 13, as follows: <<As shown in Koh *et al.*, 2015 (Ref. 21), in control experiments for the optimized ribose-seq protocol, we found that exclusion of either AtRNL or alkali treatment prevented library formation (**Supplementary Fig. 10**)>>.

3. In the Introduction, the authors stated that the current ribose-seq technique has been substantially improved relative to the original version and referred to a methods paper that is in press. It would be beneficial to general audience if the authors could highlight the modifications they made and how they improved the quality of the data presented.

- We thank the Reviewer for the suggestion and have highlighted the specific improvements in the Supplementary Methods. As describe in above in response to Reviewer 1, we added this text in the Suppl. Methods on page 31, line 9: <<We optimized the ribose-seq protocol by i) redesigning the molecular barcode-containing adaptor, making it shorter and removing overlapping sequences, ii) fragmenting the genome of interest in smaller fragments (~450 bp); iii) performing two rounds of PCR and overall reducing the PCR cycle number; iv) cutting and purifying a specific size-range of the ribose-seq library from the nondenaturing gel to eliminate any primer dimers formed during PCR and any long products that are not proficient for sequencing>> .

4. Yeast culture conditions (media and potential supplements and density of culture (log phase vs stationary)) may influence ribonucleotides identified in the cell population. The authors need to be more specific on these conditions in the Methods section.

- We thank the Reviewer for this comment and we added this specific information into the text of the Supplementary Methods on page 31, line 15: << *S. cerevisiae* cells were cultured in liquid rich medium (150 mL of a 250 mL glass flask) containing yeast extract, peptone and 2% (wt/vol) dextrose (YPD) for 2

d at 30 °C with shaking to reach stationary phase with a density of $\sim 10^8$ cells/ml >>.

5. It is clear from the data that the rNMP compositions are different in different species, particularly between *Saccharomyces* and *Schizosacchamyces*. It would be helpful to understand whether this difference is dictated by the difference in the rNTP and dNTP pools. The authors could consider measuring nucleotide pools in these strains.

- We thank this Reviewer and Reviewer #3 for this valuable suggestion. We measured the rNTP and dNTP pools in two *S. cerevisiae* strains (E134 and BY4742) and in *S. pombe* strain JK105 of wild-type RNase H cells. These data have been included in the manuscript in **Supplementary Figure 1** and are discussed on page 8, line 2: << Like in *S. cerevisiae* cells, the rUTP/dTTP ratio was also found to be the lowest among rATP/dATP, rCTP/dCTP and rGTP/dGTP ratios in *S. pombe* cells of strain JK105 (Supplementary Fig. 1)>> and on page 11, line 13: << Interestingly, the rATP/dATP ratio was very high in *S. pombe* compared to *S. cerevisiae* cells (Supplementary Fig. 1), possibly supporting elevated incorporation of rA in the absence of RNase H2 function in nuclear DNA of *S. pombe*>> .

6. In Figures 1b and 4b, after normalization to background, the expected mononucleotide frequency should be 25% for each. Thus, it is misleading to label the background frequency next to the normalized data. It may be more intuitive to move these numbers next to the upper panel. It also seems inappropriate to only indicate the *S. cerevisiae* genome nucleotide composition. The same is true for the dinucleotide heatmaps.

- We agree with the Reviewer and we moved the % values to the upper panel in Figure 1b and 4b. In these figures, we indicate the *S. cerevisiae* nucleotide compositions because *S. cerevisiae* strains represent most strains we used, and there is no space in the figures to add the corresponding % values for *S. paradoxus* and *S. pombe*. We do indicate in the legend of Figures 1b and 4b that all these values for all the 3 yeast species are shown in Supplementary Table 4: << The actual percentage of A, C, G and T bases present in mitochondrial DNA of *S. cerevisiae* are shown to the left of the bottom heatmap, and are indicated in Supplementary Table 4A. The corresponding base percentage for mitochondrial DNA of *S. paradoxus* and *S. pombe* are indicated in Supplementary Table 4B and 4C, respectively>>. The same was done for the dinucleotide % values. For better clarity, we added the text <<*S. cerevisiae* % value >> underneath the % values shown in Figure 1b and 4b. The same was done for the dinucleotide heatmaps of Figure 3a and 3b and **Supplementary Figures 3A-I, 4A-J and 6A-I**. We also ensured that all Suppl. Tables were properly uploaded and accessible to the Reviewers.

7. "These results suggest that rGs tend to be hotspot sites of rNMP incorporation in mitochondrial DNA of wild-type RNase H cells." I think the authors meant "rC" in this sentence.

- The statement in our manuscript is correct. << We further observed that while proportions of incorporated rC and rG were similar in most libraries, rC was incorporated at a larger number of different sites than rG (Fig. 1d and Supplementary Fig. 1A). These results suggest that rGs tend to be hotspot sites of rNMP incorporation in mitochondrial DNA of wild-type RNase H cells >>. If there is similar number of reads with rCs and rGs, and rCs are incorporated at a larger number of sites, it means rCs are more spread in the genome, while rGs are concentrated in specific sites, hotspot sites. To clarify this point that appeared confusing, we edited this text on page 4, line 20 as follows: << We further observed that while proportions of incorporated rC and rG were similar in most libraries, rC was incorporated at a larger number of different sites than rG (Fig. 1d and Supplementary Fig. 1A). These results suggest that rCs are

more widely spread than rGs, and thus, rGs tend to be concentrated in a smaller number of sites in mitochondrial DNA of wild-type RNase H cells >>.

8. Figure 1d-i. It is unclear what the y-axis means. The frequency exceeds 1 at some positions.

- The y-axis for Figure 1d-i (log Normalized Frequency) shows the frequency of each type of nucleotide present in the ribose-seq data normalized to the frequency of the corresponding nucleotide present in the reference genome. We calculated the raw frequency of each type of nucleotide and then divided those frequencies by the frequencies of the corresponding nucleotides in the reference genome. Therefore, since these are normalized frequencies, it is possible that the frequencies may exceed 1 at any given position. If the frequencies are greater than 1, then the frequencies are greater than what would be expected based on the nucleotide content of the reference genome. We modified the text in legend of Figure 1d-i, as well as in all Figures showing these plots (Figure 2, Figure 4d-i, and Supplementary Figures 1A-H and 5A-H): << Position 0 on the x-axis represents the site of rNMP incorporation, - and + positions represent upstream and downstream dNMPs, respectively. The y-axis shows the frequency of each type of nucleotide present in the ribose-seq data normalized to the frequency of the corresponding nucleotide present in the reference genome of the indicted yeast species>>. We also added the following explanation in the Supplementary Methods in the section ‘Nucleotide sequence context of embedded rNMPs’: << ... The Sequence Module normalizes the nucleotide frequencies to the frequencies of the corresponding reference genome. To normalize the nucleotide frequencies, the number of each type of nucleotide (A, C, G, U/T) present in the ribose-seq data was counted and divided by the total number of number of nucleotides at a given position to yield the raw proportion. In addition, the number of each type of nucleotide (A, C, G, T) present in the reference genome was counted and divided by the total number of nucleotides in the reference genome to yield the reference proportion. Then, the raw proportions were divided by the corresponding reference proportions to yield the normalized nucleotide frequencies>>

9. The criteria for “common rNMP sites” need to be better defined. Is the lack of “common sites” in the nuclear genome libraries due to the lower sequence coverage in comparison with mitochondria libraries?

- We thank the Reviewer for the comment. The term ‘common’ means rNMP sites that are in common or shared among different libraries. We modified the following text containing the term ‘common’: page 12, from line 7 <<The ribose-seq data analysis revealed rNMP sites that were in common among libraries... We generated a list of rNMP sites that were shared among all the different mitochondrial or nuclear libraries of wild-type or *rnh201*-null *S. cerevisiae* cells... Interestingly, the most abundant and shared rNMP sites in the mitochondrial libraries... Moreover, the majority of these common sites were ArC or ArG in mitochondrial wild-type (8/11 shared sites)... No shared sites were found among nuclear libraries of wild-type *S. cerevisiae*... Among the nuclear shared sites in *rnh201*-null cells...>>.

10. The authors selected top 100 rNMP sites for motif analysis. This seems arbitrary. Why not use all identified rNMP sites?

- We thank the Reviewer for pointing this out because we incorrectly reported that we selected the top 100 rNMPs. In actuality, we selected the top 1% from each library. On page 12, from line 17, we corrected to: <<To determine whether there were overlapping features and specific signatures among rNMPs that were most frequently incorporated in mitochondrial and nuclear DNA, we selected the top 1% most abundant rNMP sites from each mitochondrial and nuclear library and analyzed them using the MEME program>>. We added the number of rNMP sites for each library in Figure 5 under each MEME

plot. The legend of Figure 5a,b was modified to <<(a) Sequence motif plots for mitochondrial and (b) nuclear *rnh201*-null hotspots (top 1% of rNMP sites). The position ‘0’ corresponds to the rNMP. Each library name is indicated below each plot with its corresponding strain name, genotype, restriction-enzyme (RE) set used, and the number of rNMP sites>>. In addition, we also selected the top 100 most abundant rNMP sites, see page 12, line 22: << For comparison, we also selected the top 100 most abundant rNMP sites. The results of both analyses revealed specific consensus motifs of rNMP incorporation for these hotspot sites in mitochondrial (Fig. 5a, **Suppl Fig. 7A**) and in nuclear DNA only for *rnh201*-null libraries (Fig. 5b, **Suppl Fig. 7B**). >>, these data are included in **Supplementary Fig. 7A,B** (and **Supplementary Fig. 8B** for the emRiboSeq data). See also Methods section on ‘Genome browser and hotspots’: << Top 1% and top 100 most abundant rNMP sites were calculated using custom scripts based on the BED files created by the Coordinate Module of Ribose-Map>>. We did not use all identified rNMPs for the motif analysis because we were interested in analyzing the most abundant sites with rNMPs in each library, and because data with all rNMP sites are already presented in the nucleotide plots (Figures 1d-i, Figure 2, Figure 4d-i, Supplementary Figures 1 and 5) as well as in the heatmaps (Figs. 1b, 3, 4b, Suppl. Figs. 3A-I, 4A-J and 6A-I). The focus here is to identify the most abundant sites in every library and compare these with each other.

11. In Figures 5 c, d, the authors show examples of individual rNMP hotspots. It would be important to show that these potential hotspots are not due to certain artifacts not related to rNMPs. Thus, preparing libraries without alkali treatment seems necessary.

- We do agree that it is important control to prepare ribose-seq libraries w/o alkali treatment. In fact, this is exactly what we first did in our published study by Koh *et al.*, *Nat. Methods* 2015. In Supplementary Fig. 3 of Koh *et al* 2015, we clearly showed that w/o alkali treatment we do not obtain a detectable ribose-seq library (we note that while high pH strongly promotes hydrolysis at rNMP sites in DNA, such hydrolysis can occur, to a much lesser extent, also in the absence of specific alkali treatment of DNA). We show this same Figure here below as **Figure R3b**. We also added another example from a ribose-seq library prepared with our optimized ribose-seq protocol w/o alkali treatment, and we did not detect the library (**Supplementary Figure 10**, in the revised manuscript).

In addition, the results shown in Figures 5c,d of our manuscript, like all results, were obtained after elimination of all PCR duplicates. As explained in the Supplementary Methods in the section on “Processing and alignment of sequencing reads”, our ribose-seq adaptors contain a Unique Molecular Identifies (UMI) with 8 random nucleotides: << The UMI corresponds to sequencing cycles 1-6 and 10-11, the molecular barcode corresponds to cycles 7-9, the tagged nucleotide corresponds to cycle 12, and the tagged nucleotide’s downstream sequence corresponds to cycles 13+ of the raw FASTQ sequences. ...The Alignment Module de-multiplexed the trimmed reads by the appropriate molecular barcode, aligned the reads to the reference genome using Bowtie 2, and de-duplicated the aligned reads using UMI-tools>>. Thus, all reads that are analyzed are not PCR duplicates because they have different UMI. For the hotspots in the short nucleotide repeat tracts, in the text of the manuscript, in the legend of Figure 5d we wrote: <<All reads shown here have distinct UMI, meaning they do not represent PCR duplicates caused by slippage of DNA polymerase>>, and similarly, as in the Discussion section on page 15, line 7 from the bottom we stated: << Because practically all reads found in the short-nucleotide repeat regions had unique UMI, and the patterns were observed across multiple libraries, the repeat tracts represent hotspot sites of rNMP incorporation in yeast DNA>>. We meant that all reads in a given repeated region have different UMI, not just those reads that are at the same position in the repeat. We made this clearer in the Results by adding this text on page 13, line 22: << These are not artifacts of PCR because all reads

in a given short-nucleotide repeated tract region have different UMI, not just those reads that are at the same position within the repeated tract>>. An additional note is that for rNMP sites found in short-nucleotide repeated tracts, we only considered those reads that had longer sequence than the repeats. In the Supplementary Methods, we added this text on page 38, line 12: << For the analysis of short nucleotide repeat tracts, we only considered reads that were longer than the repeated regions to ensure accuracy of our findings >>.

12. “Interestingly, we found rG to be present below the standard level of G in all nuclear wild-type libraries of *S. cerevisiae*, *S. paradoxus* and *S. pombe*.” Define “standard level”.

- For better clarity, we edited this text as follows: << Interestingly, we found rG to be present below the proportion of dGMP in all nuclear wild-type libraries of *S. cerevisiae*, *S. paradoxus* and *S. pombe* >>. Moreover, we defined more clearly ‘standard frequency’ the first time this is mentioned in the text on page 5, from line 10: << We reasoned that, if for example rA was randomly incorporated in mitochondrial DNA of *S. cerevisiae*, the frequency by which the dNMP A, C, G or T was found at position -1 and +1 relative to rA should reflect the frequency by which the dinucleotides AA, CA, GA, TA, AC, AG and AT occur in mitochondrial DNA of *S. cerevisiae* (standard frequency)>>.

Reviewer #3 (Remarks to the Author):

The Storici lab developed one of three techniques that can be used to map the identities/positions of rNMPs embedded in genomic DNA and ribose-seq is the only technique that eliminates the background noise of DNA damage. In the current study, this technique is used to map rNMPs in multiple *S. cerevisiae* strains, an *S. paradoxus* strain and an *S. pombe* strain (close and distant relatives of *S. cerevisiae*, respectively). Both WT and RNase H2-defective backgrounds were used, as well as several additional *S. cerevisiae* strains (RED, a PIP-defective mutant RNase H2 mutant, and an *rnh1* mutant). This was a Herculean effort that generated a data-rich manuscript, and also may be the first example of mapping nuclear rNMPs in a WT background. rNMPs were examined in mtDNA, and data confirm the lack of RER in mitochondria and demonstrate a strong over-representation of rC in *S. cerevisiae* and *S. paradoxus* but rG in *S. pombe* (Figure 1). An additional difference is a bias for rNMP insertion in CG-rich regions in *S. cerevisiae* and *S. paradoxus*, but not in *S. pombe* (Figure 2). Finally, there is a clear effect of the previously incorporated dNMP on rNMP incorporation/identity (Figure 3). In terms of rNMP patterns/levels in nuclear DNA, the most striking results are the difference between WT and *rnh201* mutants (Figure 4) and the patterns of incorporation in repetitive DNA (Figure 5B). There are multiple important insights provided by the analyses, and only minor revisions are suggested.

- We thank the Reviewer very much for her/his comments and for valuing the findings of our study.

1. There needs to be some comment in the main text concerning the choice of *S. cerevisiae* strains used.

- We agree, we added the following text at the beginning of the Results on page 4, from line 1: <<We built and analyzed 11 ribose-seq libraries with mitochondrial DNA of wild-type RNase H2 cells from six among the most commonly utilized haploid yeast lab strains (see Supplementary Methods) using 3 sets of

restriction enzymes (RE): E134 (RE1, RE3), BY4741 (RE1), BY4742 (RE1, RE2, RE3), YFP17 (RE1, RE2, RE3), W303 (RE3) and S288C (RE2) (Supplementary Tables 1-4)>>.

2. If the authors are going to present formulas in Figures 1 and 4, they should provide some explanation in the main text. The normalized values will obviously be affected by concentrations in the corresponding rNTP pool. How will this affect the over- or under-representation of rNMPs?

- In the Results we had following text: <<Normalization of single rNMP frequencies to the nucleotide base content of *S. cerevisiae* mitochondrial DNA revealed marked bias for rC and/or rG incorporation in all libraries over rA and especially over rU (Fig. 1b,c,...>>, and we edited the text further on (page 9, line 1): << In wild-type RNase H2 cells, rG was consistently the least abundant rNMP in all the libraries, and normalization of single rNMP frequencies to the nucleotide base content of *S. cerevisiae* nuclear DNA revealed marked bias for rC (Fig. 4a-c...>>. Moreover, we agree that the variation in rNTP and dNTP pools may be responsible for variations in rNMP patterns among different strains and species. As also suggested by Reviewer #2, we measured rNTP and dNTP pools in *S. cerevisiae* strains E134 and BY4742, as well as in *S. pombe* (Supplementary Figure 1). Our analysis showed that the rGTP/dGTP ratio is higher in BY4742 vs E134 supporting higher incorporation of rG in BY4742 compared to E134. Moreover, we observed that the rUTP/dTTP ratio is the lowest of the rNTP/dNTP ratios, not only in the tested strains of *S. cerevisiae*, but also in the strain of *S. pombe*. This result suggests that rU incorporation might be the least favored among all rNTPs. It is consistent with rU being the least abundant rNMP in mitochondrial DNA and in nuclear DNA of *rnh201*-null cells of both *S. cerevisiae* and *S. pombe*. We also observed a high ratio rATP/dATP in *S. pombe*, much more than in *S. cerevisiae* cells. It is possible that this high ratio facilitates the incorporation of rA in nuclear DNA of *S. pombe* when the *RNH201* gene is knocked out. These results are discussed on page 4, from line 18: <<By measuring rNTPs/dNTPs ratios for strains E134 and BY4742, we found that the rGTP/dGTP ratio was significantly lower in E134 compared to BY4742 ($P = 0.0008$) (Supplementary Figure 1), possibly accounting for the lower rG incorporation in E134>>, on page 8, line 2: << Like in *S. cerevisiae* cells, the rUTP/dTTP ratio was also found to be the lowest among rATP/dATP, rCTP/dCTP and rGTP/dGTP ratios in *S. pombe* cells of strain JK105 (Supplementary Fig. 1)>>, and on page 11, line 13: << Interestingly, the rATP/dATP ratio was very high in *S. pombe* compared to *S. cerevisiae* cells (Supplementary Fig. 1), possibly supporting elevated incorporation of rA in the absence of RNase H2 function in nuclear DNA of *S. pombe*.>>, respectively.

3. A bit more discussion of possible RNase H2 preferences would be helpful.

- A just published study by Lietard et al., *Biochemistry* 2019 ‘Large-scale photolithographic synthesis of chimeric DNA/RNA hairpin microarrays to explore sequence specificity landscapes of RNase HII cleavage’ (DOI: 10.1021/acs.biochem.9b00806) summarizes previous evidence showing cleavage preference by RNase H enzymes, and demonstrates clear substrate preference for RNase HII of *E. coli*. The cleavage preference is localized around and at the cleavage site. In the Discussion, we edited the following text on page 16, line 15: << Therefore, the fact that wild-type RNase H2 cells have low rG and equal absolute amount of rA, rC and rU in nuclear DNA, with an overall rNMP distribution that is significantly different from that observed in *rnh201*-null cells (Fig. 4b,c and Fig. 1b,c), may indicate that yeast RNase H2 cleaves rNMPs in yeast DNA with differential efficiency and may have slight preference for rG, as preliminary data showed using protein extracts from yeast, HeLa cells and *Escherichia coli* (Rydberg and Game, 2002). A more recent study, exploiting microarray analysis of thousands of rNMP-containing DNA hairpins of different sequence, demonstrated clear cleavage preference of *E. coli* RNase HII at and around the cleavage site (Lietard et al., *Biochemistry* 2019)... >>.

4. Perhaps the under-representation of U reflects the preference of Top1 to incise at T?

- It is possible that underrepresentation of rU is affected by multiple factors, nt pools, proofreading and Top1 activity. The fraction of rU incorporation was increased in the *rmh201*-null *pol2-4* mutant of *S. cerevisiae* as we showed in Koh *et al.*, *Nat. Methods* 2015 in Fig. 3j. The analysis of rNTP and dNTP pools that we conducted for two *S. cerevisiae* and 1 *S. pombe* strains shows that the rUTP/dTTP ratio is always the lowest of the rNTP/dNTP ratio both in *S. cerevisiae* and *S. pombe* cells. These results are presented in Supplementary Fig. 1 and are discussed in the text on page 8, line 2 << Like in *S. cerevisiae* cells, the rUTP/dTTP ratio was also found to be the lowest among rATP/dATP, rCTP/dCTP and rGTP/dGTP ratios in *S. pombe* cells of strain JK105 (Supplementary Fig. 1)>> and on page 16, line 2: << While the low level of rU can be explained by general high concentration of dTTP in the nucleotide pools (Clausen *et al.*, 2013; Wanrooij *et al.*, 2017), with rUTP/dTTP being the lowest rNTP/dNTP ratio, which we observed both in *S. cerevisiae* and *S. pombe* strains (Supplementary Fig. 1), potential activity of topoisomerase I on sequences with rU (Klein, 2017; Cho *et al.*, 2018), and some proofreading activity for Pol ϵ on rU (Koh *et al.*, 2015)...>>.

5. Any ideas why there is a repetitive pattern of rNMPs in repeated sequence?

- We think that specific sequence contexts favor incorporation of rNMPs. Our analyses of rNMP sites via heatmaps and the MEME program revealed that it is the sequence of the rNMPs and that of the nucleotide immediately upstream of the rNMPs that are most conserved. For example, the dinucleotides AC and AG were found to be ‘hotspots’ of rNMPs as ArC and ArG in the yeast genome. Thus, if the same dinucleotide-sequence context is repeated within a given region of the genome, like in tri- or di-nucleotide repeated tracts containing such dinucleotides, any AC or AG site within the repeated tracts may be equally prone to rNMP incorporation. Our libraries derive from a population of cells, not from a single cell; thus, the pattern we observe in the repeated tracts highlights what is present in a population of cells. In the cell population any unit within the short-nucleotide repeated tracts containing AC or AG nucleotides is prone to rNMP incorporation. We added the following text in the Results section specific to repetitive patterns of rNMPs on page 17, line 7: <<Remarkably, in all repeats that we analyzed except one (9/10), the rCs and rGs were preceded by dA (Fig. 5c,d and Supplementary Fig. 7). Thus, if the same dinucleotide-sequence context with AC or AG is repeated within a given region of the genome, like in tri- or di-nucleotide repeated tracts containing such dinucleotides, any AC or AG site within the repeated tracts may be equally prone to rNMP incorporation, explaining the observed patterns >>.

Minor issues

“Differently” is awkwardly used throughout.

- We checked all times we use the term “differently” throughout the manuscript, identified and edited those that appeared awkward. Specifically, the term “Differently” was removed from the sentence on page 7, line 23: << The pattern of rNMP incorporation in mitochondrial DNA of *S. pombe* displayed a stronger preference for rG... >>, successive statements were modified to: << The two *RED* nuclear libraries, while still showing rC more prevalent and rU as least abundant, had increased level of rG>> on page 9, line 13; << *S. pombe* showed ArA as dominant motif in *rmh201*-null cells>> on page 13, line 15, and << We found more consistent rNMP distribution in nuclear DNA among all the strains for wild-type RNase H2 cells of *S. cerevisiae*...>> on page 15, line 24.

On the middle of p. 14, should the reference be to Figures 1B and 4B instead of 1C and 4C, which are formulas?

- Correct, we thank the Reviewer. We corrected to: << E134 displayed marked preference for rC versus rG in mitochondrial DNA (Fig. 1b) >> on page 15, line 21, and to: << We found more consistent rNMP distribution in nuclear DNA among all the strains for wild-type RNase H2 cells of *S. cerevisiae*, and similarly for *rnh201*-null cells (Fig. 4b)>> on page 15, line 24.

Middle of p. 15 – motifs, not motives

- Correct, we thank the Reviewer. We corrected to: << Strikingly, ArC and ArG were dominant motifs in all hotspots... >> on page 17, line 3.

Figure R1

a

rNMP Composition in Mitochondria

b

rNMP Composition in Nucleus

***S. pombe rnh201*-null DNA**

Bar graphs with normalized rNMP composition for 3 *S. pombe rnh201*-null libraries prepared by Zhou Z, Burkholder AB, Lujan SA, Kunkel TA and available at this link:

<https://www.ncbi.nlm.nih.gov/geo/query/acc.cgi?acc=GSE125855>

[REDACTED]

Koh *et al.*, *Nat. Methods* 2015, Fig. 1a

[REDACTED]

[REDACTED]

Balachander & Yang *et al.*, *Meth. Mol. Biol.*, Figs. 5 and 6

Reviewers' comments:

Reviewer #2 (Remarks to the Author):

The revisions significantly improved the manuscript. However, I do not believe the fundamental concerns about this study have been satisfyingly addressed.

Throughout the rebuttal letter, the authors argue that the most significant and novel finding in this study is that the "DNA sequence context shapes the pattern and distribution of rNMPs in genomic DNA" and claim this observation to be "unexpected". I don't think I can agree with the authors on how "unexpected" the observation is. On the contrary, with some knowledge on base stacking and base pairing, it is quite expected that DNA sequence context may play a role in ribonucleotide incorporation, i.e. some sequence contexts are more accommodating to rNMP. Analogously, for instance, many studies have shown that sequence context plays a role in shaping base-pair substitutions. Detailed analysis of the sequence contexts is certainly appreciated. My concern is that all the descriptive observations do not significantly advance how the understanding of the mechanisms.

The comparison with data generated by emRibo-seq is helpful in evaluating the validity of the results presented. In my opinion, the value of the comparison is inherently limited and it does not produce sufficient independent confirmation of the authors' claim. First of all, the emRibo-seq was generated in a different *S. cerevisiae* background. If we learn anything from this study, the strain background is one of the determinants of rNMP incorporations, partly due to differences in nucleotide pools. Secondly, I don't know how the authors can conclude "good similarity" between data from the two methods, and in some cases, "identical". To, the two datasets in comparison are very different. For instance, for normalized mononucleotide frequencies in mitochondria, rG is highly preferred in emRibo-seq data (Sup. Fig. 4A) while rC is preferred in ribo-seq data. Similarly, in nuclear genome, rC is highly preferred in ribo-seq while rC and rG are equally preferred in emRibo-seq in *rnh201* mutants. Dinucleotide frequencies are also quite different. For instance, the influence of the -1 nucleotide is very modest in emRibo-seq, with the biggest difference below 2 fold (between dC and dT) in *rnh201* mutants. However, in the ribo-seq data, the differences between dC and dT at -1 positions are as high as 5 fold. Since the dinucleotide analyzed should be less influenced by nt pools, these differences likely reflect respective biases in different methods. Thus, this comparison does not alleviate any concerns regarding intrinsic biases but in fact highlights such concerns. In my opinion, emRibo-seq data do provide support to the notion that sequence contexts influence the rNMP incorporation/retention, but not much beyond that.

Related to the comment above. As I noted in my previous comments, the difference of *S. pombe* data between Daigaku et al. 2015 and the present study is quite obvious. The authors brought up another study using the HydEn-seq approach in the *S. pombe* strains. However, the profile from the HydEn-seq data is still very different from the ribo-seq. The authors cannot cherry-pick the data point to claim support (rU). To my knowledge, *S. pombe* lab strains all derive from the same origin unlike *S. cerevisiae*. All these differences again highlight possible intrinsic biases of these methods. Sequencing based methods alone may not be sufficient to solve the problem.

Reviewer 1 raised questions about the lack of replicates of individual strains. I think this statement was at least partly true. There is no replicate for some strains or genotypes: e.g. no replicate for BY4741-WT, W303-WT, S288c-WT or *S. paradoxus-rnh201*.

Reviewer 1's comments on Figure 1a, b and 4a, b have been published. I think it was pretty clear that the reviewer meant that previous studies have published rNMP contents in various *S. cerevisiae* and *S. pombe* strains. Authors should not misinterpret the reviewer's comment into suggesting that the exact same datasets have been published. I do agree with reviewer 1 on this point. As I commented before, some of the topline conclusions on rNMP identity can readily be drawn from

the published data, such as some rNMPs are preferred over others. But I agree with the authors no studies have compared these specific *S. cerevisiae*, *S. paradoxus* and *S. pombe* strains side-by-side.

Related to my previous comments on background signals. Without proper controls, authors cannot claim the individual hotspots are indeed rNMPs. The inability to prepare library w/o alkali treatment simply means that there is not enough starting material (not surprising). It does not necessarily mean that there is no background. The authors noted that the restriction sites may be captured as background noise. They should be able to estimate the extent of restriction sites captured by ribose-seq which would at least provide some idea on whether or not naturally occurring DNA break hotspots either in vivo or during DNA isolated may be misinterpreted as rNMP hotspots.

Minor comments:

Labeling of either supplementary 4E or 4F is wrong, as both were shown to analyze the +1 context.

The authors did not address my previous questions: "Is the lack of "common sites" in the nuclear genome libraries due to the lower sequence coverage in comparison with mitochondria libraries?" The "common rNMP sites" will not be biologically meaningful, if this is merely due to differences in coverage.

December 19th, 2019

Point-by-point Rebuttal, NCOMMS-19-18835A-Z

- Text of Rebuttal, Figures comparing side by side ribose-seq and emRiboSeq results, and Figures and explanations addressing the ‘rU’ contradiction between Daigaku et al., *NSMB* 2015 and our study.

Point-by-point rebuttal to Reviewer #2

The revisions significantly improved the manuscript. However, I do not believe the fundamental concerns about this studied have been satisfyingly addressed.

- We disagree with this statement of the Reviewer #2 as explained in the points below.

1) Throughout the rebuttal letter, the authors argue that the most significant and novel finding in this study is that the “DNA sequence context shapes the pattern and distribution of rNMPs in genomic DNA” and claim this observation to be “unexpected”. I don’t think I can agree with the authors on how “unexpected” the observation is. On the contrary, with some knowledge on base stacking and base pairing, it is quite expected that DNA sequence context may play a role in ribonucleotide incorporation, ie. some sequence contexts are more accommodating to rNMP. Analogously, for instance, many studies have shown that sequence context plays a role in shaping base-pair substitutions. Detailed analysis of the sequence contexts is certainly appreciated. My concern is that all the descriptive observations do not significantly advance how the understanding of the mechanisms.

- The word “unexpected” is never used in our manuscript text; thus, the Reviewer’s argument is misleading. We used “unexpected” in the Rebuttal, but only there and not in the manuscript. We indeed did not know what to expect when we conducted these experiments, and we used this term to underscore that our results are novel to the field of ribonucleotide incorporation in DNA; there is no published mention about such expectation. We further note that bridging the gap between expecting something and actually proving it requires enormous effort. Thus, writing that our results are expected and thus of little value would have been nonsense. Contrary to what the Reviewer writes, our results do significantly and markedly advance our understanding about the process of ribonucleotide incorporation in DNA, and this is exactly the crux of our paper. To reach this conclusion it took huge <<herculean>> efforts, as Reviewer #3 wrote. We believe this Reviewer’s point does not support rejection of our manuscript.

2) The comparison with data generated by emRibo-seq is helpful in evaluating the validity of the results presented. In my opinion, the value of the comparison is inherently limited and it does not produce sufficient independent confirmation the authors claim. First of all, the emRibo-seq was generated in a different *S. cerevisiae* background. If we learn anything from this study, the strain background is one of the determinants of rNMP incorporations, partly due to differences in nucleotide pools. Secondly, I don’t know how the authors can conclude “good similarity” between

data from the two methods, and in some cases, “identical”. To, the two datasets in comparison are very different. For instance, for normalized mononucleotide frequencies in mitochondria, rG is highly preferred in emRibo-seq data (Sup. Fig. 4A) while rC is preferred in ribose-seq data. Similarly, in nuclear genome, rC is highly preferred in ribose-seq while rC and rG are equally preferred in emRibo-seq in rnh201 mutants. Dinucleotide frequencies are also quite different. For instance, the influence of the -1 nucleotide is very modest in emRibo-seq, with biggest difference below 2 fold (between dC and dT) in rnh201 mutants. However, in the ribo-seq data, the differences between dC and dT at -1 positions are as high as 5 fold. Since the dinucleotide analyzed should be less influenced by nt pools, these differences likely reflect respective biases in different methods. Thus, this comparison does not alleviate any concerns regarding intrinsic biases but in fact highlights such concerns. In my opinion, emRibo-seq data do provide support to the notion that sequence contexts influence the rNMP incorporation/retention, but not much beyond that.

- This point of Reviewer #2 contains a series of inaccurate and wrong statements. <<The comparison with data generated by emRibo-seq is helpful in evaluating the validity of the results presented >>. We thank the Reviewer for this note. We highlight that this was the major criticism of this Reviewer in the first round of reviews: << without further validation, it is unclear whether the sequence preferences, which is the central focus in this study, is due to bias in ribonucleotide incorporation or bias of the approach, or perhaps more likely, a combination of both>>. We provided the requested validation by analyzing the emRiboSeq data of Reijns et al., *Nature* 2015. EmRiboSeq is the current rNMP mapping approach that is most different from ribose-seq, because it does not fragment DNA by restriction enzymes, but by sonication, does not use alkali, and does not use AtRNL, but utilizes human recombinant RNase H2 and T4 quick ligase to cleave at rNMPs and capture the broken DNA ends upstream from the rNMPs, respectively. As described in our revised manuscript (and our initial rebuttal), the analyses of the emRiboSeq data revealed striking similarity to our ribose-seq data, supporting the central focus of our study: the dNTP immediately upstream (at position -1) of the rNMP in DNA has the most impact on the pattern of rNMP incorporation (e.g. dA upstream of rC and rG).

Reviewer #2 writes << the value of the comparison is inherently limited, and it does not produce sufficient independent confirmation the authors claim>>. This is not correct. Central figures are Figure 3a,b for the *S. cerevisiae* data to be compared with emRiboSeq *S. cerevisiae* Supplementary Figure 4C,D, respectively (see these Figures attached to this Rebuttal); and Figure 5a, Suppl. Fig.7A and Figure 5d to be compared with emRiboSeq Supplementary Figure 8A, B and C, respectively (see these Figures attached to this Rebuttal). These ribose-seq and emRiboSeq figures are attached side by side for direct comparison. These results show high similarity between the ribose-seq and emRiboSeq data. We believe that this similarity is quite evident. The conclusions taken from each set of data are identical. Thus, the two sets of data lead to the same conclusion: the dNTP immediately upstream (at position -1) of the rNMP in DNA has the most impact on the pattern of rNMP incorporation (e.g. dA upstream of rC and rG). This conclusion stands independently from the *S. cerevisiae* strain used and independently from the technique (ribose-seq vs emRiboSeq) used. Thus, writing that <<First of all, the emRibo-seq was generated in a different *S. cerevisiae* background. If we learn anything from this study, the strain background is one of the determinants of rNMP incorporations, partly due to differences in nucleotide pools.>> is wrong, it denies the evidence.

The statement << Secondly, I don't know how the authors can conclude “good similarity” between data from the two methods, and in some cases, “identical”. To, the two datasets in comparison are

very different.>> is wrong, see attached figure comparisons (Figure 3a compared to Figure 4C; Figure 3b compared to Suppl. Figure 4D, and Figure 5a, Suppl. Fig. 7A and Figure 5d to be compared with emRiboSeq Supplementary Figure 8A-C), which are the central focus of this study, as discussed above.

The statement << For instance, for normalized mononucleotide frequencies in mitochondria, rG is highly preferred in emRibo-seq data (Sup. Fig. 4A) while rC is preferred in ribose-seq data. Similarly, in nuclear genome, rC is highly preferred in ribose-seq while rC and rG are equally preferred in emRibo-seq in rnh201 mutants. >> is all correct, what is wrong is the interpretation that the Reviewer makes of these data. In fact, it is exactly what we state in the first section of Results, that there are differences at the mononucleotide level, see our Figure 1b and the text on page 4, line 6: <<we revealed some variation among the strains (Fig 1a,b; Supplementary Table 3)>>. The mononucleotide heatmaps show high rG in mitochondrial DNA for the libraries of emRiboSeq, and in the rebuttal we stated: “For the mononucleotides, background Δ I(-2)I-7BYUNI300 of the emRiboSeq libraries has more similarity to the BY4742 strain that we used in ribose-seq libraries, with rC and rG prevalent and rU as the lowest after normalization in mitochondrial DNA, and also higher rC and rG and lowest rU for nuclear DNA (**Suppl. Figure 4A,B**).” This is all in line with what we state. There are strain differences at the mononucleotide level, but what is highly conserved is the pattern at the dinucleotide level, which Reviewer #2 totally missed.

Reviewer #2 then writes: << Dinucleotide frequencies are also quite different. For instance, the influence of the -1 nucleotide is very modest in emRibo-seq, with biggest difference below 2 fold (between dC and dT) in rnh201 mutants. However, in the ribo-seq data, the differences between dC and dT at -1 positions are as high as 5 fold. Since the dinucleotide analyzed should be less influenced by nt pools, these differences likely reflect respective biases in different methods. Thus, this comparison does not alleviate any concerns regarding intrinsic biases but in fact highlights such concerns. In my opinion, emRiboSeq data do provide support to the notion that sequence contexts influence the rNMP incorporation/retention, but not much beyond that>>. All this part is wrong. As stated above, there is strong similarity between ribose-seq and emRiboSeq for -1 data. Please, compare mitochondrial data of Fig. 3a with Suppl. Figure 4C (attached), these data give identical conclusions. The Reviewer never provides any comment on all these -1 data. We do not know what figures the Reviewer was looking at, but the Reviewer’s conclusion is wrong. Reviewer #2 does not list figure numbers here, showing that the Reviewer’s work was superficial. Also, for nuclear data there is good similarity between ribose-seq and emRiboSeq data, see Figure 3b and Suppl. Figure 4D (attached). Although less strongly than for mitochondrial data, these nuclear data of emRiboSeq also give the same conclusion as ribose-seq data. We note that we made and provide statistical analysis of all the ribose-seq heatmap data, these are shown in Supplementary Table 5. In addition, comparison of data shown in Figure 5a with Suppl. Fig. 8A; and Suppl. Fig. 7A with Suppl. Fig. 8B; and Figure 5d with Suppl. Fig. 8C (all attached) support the same conclusions from these sets of data, that the -1 nucleotide is most relevant in rNMP incorporation. Again, these marked similarities between ribose-seq and emRiboSeq data were never considered by this Reviewer.

Overall the above points of the Reviewer do not support rejection.

3) Related to the comment above. As I noted in my previous comments, the difference of *S. pombe* data between Daigaku et al. 2015 and the present study is quite obvious. The authors brought up another study using the HydEn-seq approach in the *S. pombe* strains. However, the profile from the HydEn-seq data is still very different from the ribose-seq. The authors cannot cherry-pick the data point to claim support (rU). To my knowledge, *S. pombe* lab strains all derive from the same origin unlike *S. cerevisiae*. All these differences again highlight possible intrinsic biases of these methods. Sequencing based methods alone may not be sufficient to solve the problem.

- Due to the stark reproducibility of our results, we feel very strongly about our rU data for all the 3 yeast species including *S. pombe* for the *rnh201*-null genotype, and the same is for mitochondrial data of all 3 yeast species in wild-type and *rnh201*-null cells. Our rU data are markedly reproducible in our study, and we also reproduced them using the data and libraries of Zhou et al. 2019 that the *Nat. Commun.* journal recently published.

Reviewer #2 also incorrectly cites the <<HydEn-seq approach >> as the technique that was used in Zhou et al., *Nat. Commun.* 2019. This is not trivial for experts in the field; the technique is actually called RHII-HydEn-seq, which is completely different from HydEn-seq (this is described in the Zhou et al. paper; they changed the method and name from HydEn-seq to RHII-HydEn-seq). This incorrect terminology exposes may be rushed and/or light analysis by Reviewer #2 in the review of our manuscript and the revisions made.

Reviewer #2's statement: <<However, the profile from the HydEn-seq data is still very different from the ribose-seq. The authors cannot cherry-pick the data point to claim support (rU)>> is misleading and in part offensive. Reviewer #2's critique in the first round of reviews was about the contradictory results of rU in *S. pombe*, and this is what we responded to, to rU. We do not make a strong point about the most abundant rNMP in *S. pombe*, but we do make a strong point that all 3 yeast species have low rU in mitochondria, and in nucleus for *rnh201*-null cells. The term "cherry-pick" in this context is thus an inappropriate term.

Furthermore, with the aim to understand why data of Daigaku et al. *NSMB* 2015 do not show low rU at the rNMP position, we worked to address such contradiction, which is a significant concern for Reviewer #2. We believe we solved the problem. Focusing on Daigaku et al., *NSMB* 2015, on their Suppl. Figure 3 (included in our Rebuttal file), which shows the ratios of nucleotides with base A,C,G,T/U at the rNMP position (0) and from position -3 to +3 relative to the rNMP, we note that they display very low dT at the +1 position. The nuclear *S. pombe* genome is A/T-rich (nt composition is shown under this figure); thus, it is quite unlikely that position +1 relative to the rNMP would have markedly low dT content, particularly since all of the other positions do not have low dT. Based on this knowledge, we thought it would be important to investigate if the +1 position in Daigaku et al.'s Supplementary Figure 3 was accidentally shifted during computational analysis and actually corresponds to the 0 position (the position of the rNMP) in ribose-seq and in the Zhou et al.'s *Nature Communications* 2019 study, which both show low rU for the rNMP position. To evaluate if the data were shifted, we re-analyzed one of the Pu-seq *S. pombe rnh201*-null libraries shown in their Suppl. Fig. 3 (GSM1519714) using the bioinformatics code outlined in Keszthelyi et al.'s *Nature Protocols* 2015 manuscript (doi:10.1038/nprot.2015.116) also by Dr. A. Carr's group. This manuscript provides more details regarding the computational analysis required for Pu-seq data. Based on this manuscript, we obtained the results shown in **Figure R1** (see figure below), which match well with those that we have in our manuscript, with rU being low at the rNMP position. Below the figures, on the last page of the Rebuttal, we have explained in more detail what we think

might have occurred during the computational analysis for the Daigaku et al. manuscript that caused such a shift in their data.

Overall, we do not believe this point of Reviewer #2 is valid to reject our manuscript.

4) Reviewer 1 raised questions about the lack of replicate of individual strains. I think this statement was at least partly true. There is no replicate for some strains or genotypes: eg. no replicate for BY4741-WT, W303-WT, S288c-WT or *S. paradoxus*-*rnh201*.

- We clearly stated all numbers of repeats we did, this is shown in practically all of our figures and Supplementary Tables 3 and 6. All of our major findings are supported by large number of reproducible results, from 34 mitochondrial data sets and 25 nuclear data sets. In addition, we also have the 5 emRiboSeq libraries, including 5 sets of mitochondrial and 5 sets of nuclear data. *S. cerevisiae* BY4741-WT, W303-WT and S288c-WT are counted as wild-type strains to be compared with *rnh201* strains. We have 11 wild-type libraries for *S. cerevisiae* mitochondrial data and 6 wild-type libraries for *S. cerevisiae* nuclear data. We do not make strong points on any of these WT strains, but highlight some strain diversity for the mononucleotide data. For the dinucleotide data, which are the central focus of our study, all of these wild-type strains behave similarly and show a very similar pattern, see Figure 3a for mitochondrial data and 3b for nuclear data of *rnh201*-null cells. For *S. paradoxus* we were mainly interested in wild-type RNase H2, but we included one library that is from *rnh201*-null cells. This library gives the same results in mitochondria as *S. paradoxus* wild-type cells and the same results as *S. cerevisiae rnh201*-null nuclear libraries. The point of this Reviewer is out of context and forces our data in a direction that we are not taking. This is a misinterpretation of the results, which underscores that this Reviewer did not carefully analyze and understand our data of this study. Therefore, we believe this point does not support rejection.

5) Reviewer 1's comments on Figure 1a, b and 4a, b been published. I think it was pretty clear that the reviewer meant that previous studies have published rNMP contents in various *S. cerevisiae* and *S. pombe* strains. Authors should not misinterpret the reviewer's comment into suggesting that the exact same datasets have been published. I do agree with reviewer 1 on this point. As I commented before, some of the topline conclusions on rNMP identity can readily be drawn from the published data, such as some rNMPs are preferred over others. But I agree with the authors no studies have compared these specific *S. cerevisiae*, *S. paradoxus* and *S. pombe* strains side-by-side.

- We provide here what is present in Figures 1 and 4 that was previously published: nothing. In Figure 1a and 4a nothing was done before. The *S. cerevisiae rnh201* cells of strain E134 in the *rnh201*-null genotype were used in our Koh et al., *Nat. Methods* 2015 but with a different set of restriction enzymes (corresponding to RE1, which we do not use in this data set for *rnh201* cells of E134, because we use RE2 and RE3). We note that using different sets of restriction enzymes that we implement in this study is important to validate the reproducibility of results independently from the RE set used. For nuclear *S. pombe* data of *rnh201* cells, there was only the Daigaku et al., 2015 paper, for a different *S. pombe* strain, which we know to be in contradiction with our results. Figures 1b and 4b, the heatmaps of mitochondrial and nuclear rNMP frequencies, respectively, were never published before by anyone.

If there was concern about the other parts of Figure 1 and 4, we note that section (c) are formulas and none of the panels in d-i in both figures show data from strains that were published before.

We believe that the above criticisms of Reviewer #2, and #1 earlier, are unfounded and do not justify rejection of our manuscript.

6) Related to my previous comments on background signals. Without proper controls, authors cannot claim the individual hotspots are indeed rNMPs. The inability to prepare library w/o alkali treatment simply means that there is not enough starting material (not surprising). It does not necessarily mean that there is no background. The authors noted that the restriction sites may be captured as background noise. They should be able to estimate the extent of restriction sites captured by ribose-seq which would at least provide some idea on whether or not naturally occurring DNA break hotspots either in vivo or during DNA isolated may be misinterpreted as rNMP hotspots.

- In the Materials and Methods section, we clearly describe our methodology, with all controls, starting on page 35 from line 21. We clearly explain that our results are not affected by the capture of restriction enzyme sites. The top 1% and top 100 most abundant rNMP sites for the ribose-seq and emRiboSeq libraries underscore that the nucleotide at position -1 from the rNMP is most conserved in each library (Figure 5a,b and Supplementary Fig. 7A,B for ribose-seq data; and Supplementary Fig. 8A,B for emRiboSeq data). Thus, based on the 1% and top 100 sites in ribose-seq and emRiboSeq libraries, we reach the same conclusion. We note that emRiboSeq libraries were not prepared using restriction enzymes and alkali. Moreover, the rNMP sites that were common among our mitochondrial WT, *rnh201*-null, or nuclear *rnh201*-null libraries are all listed in Supplementary Table 7A,B and C, respectively. In this table, we provide names of all of libraries that have the common sites. These libraries were prepared using different sets of restriction enzymes, yet they have common sites. We also provide the sequence context of these rNMPs, and this sequence context clearly does not reveal any sites of restriction enzymes, please see Supplementary Table 7 of the manuscript.

The concern of the Reviewer in this point is also unfounded and does not justify rejection of our manuscript.

Reviewer #2 had 2 minor points.

Minor comments:

1) Labeling of either supplementary 4E or 4F is wrong, as both were shown to analyze the +1 context.

- We had correctly labelled Supplementary Figure 4E and F. In Supplementary Figure 4E we show the RN dinucleotide frequency normalized to background frequency in mitochondria; while in Supplementary Figure 4F we show the RN dinucleotide frequency normalized to background frequency in nucleus.

2) The authors did not address my previous questions: “Is the lack of “common sites” in the nuclear genome libraries due to the lower sequence coverage in comparison with mitochondria libraries?” The “common rNMP sites” will not be biologically meaningful, if this is merely due to differences in coverage.

- We apologize for missing to address this point of the Reviewer. Wild-type RNase H2 libraries generally have lower coverage. It is possible that low coverage makes it more difficult to find common sites. However, we disagree with the Reviewer’s statement that common rNMP sites are not biologically meaningful if they are more easily identified in libraries with high coverage like those of *rnh201*-null cells. Further experiments should be done to determine whether there are common rNMP sites in wild-type nuclear DNA of yeast cells. For example, the mating type switching site of *S. pombe* should be a common, biologically relevant rNMP site in wild-type RNase H2 *S. pombe* cells. In addition, there might be hotspot sites of rNMP incorporation that are more easily targeted by RNase H2, and these could be detected as common sites in *rnh201*-null cells.

Figure 3a

ribose-seq data

NR

Dinucleotide frequency normalized to background frequency in mitochondria

Suppl. Fig. 4C

Dinucleotide frequency normalized to background frequency in mitochondria

emRiboSeq data

Fig. 3b

Suppl. Fig. 4D

ribose-seq data

mitochondrial hotspots, top 1% of rNMP sites

emRiboSeq data

mitochondrial hotspots, top 1% of rNMP sites

ribose-seq data

mitochondrial hotspots, top 100 rNMP sites

Sup. Fig 7

emRiboSeq data

mitochondrial hotspots, top 100 rNMP sites

Sup. Fig 8

Figure 5

ribose-seq data

d

E134-*rnh201* RE2-FS156
ChrM 63563 - 63672

Sup. Fig 8

emRiboSeq data

C

$\Delta(-2)|-7$ BYUNI300 *rnh201* SRR1734980

ChrM: 63,563-63,672

[REDACTED]

The nuclear *S. pombe* genome is A/T-rich (nucleotide composition is shown under the figure); thus, it is quite unlikely that position +1 relative to the rNMP would have markedly low dT content, particularly since all of the other positions do not have low dT. Based on this knowledge, we thought it would be important to investigate if the +1 position in Daigaku et al.'s Supplementary Figure 3 was accidentally shifted during computational analysis and actually corresponds to the 0 position (the position of the rNMP) in ribose-seq and in the Zhou et al.'s *Nature Communications* 2019 study, which both show low rU for the rNMP position.

To evaluate if the data were shifted, we re-analyzed one of the Pu-seq *S. pombe rnh201*-null libraries shown above (GSM1519714) using the bioinformatics code outlined in Keszthelyi et al.'s *Nature Protocols* 2015 manuscript (doi:10.1038/nprot.2015.116) also by Dr. A. Carr's group. This manuscript provides more details regarding the computational analysis required for Pu-seq data. Based on this manuscript, we obtained the results shown below, which match well with those that we have in our manuscript, with rU being low at the rNMP position, see Figure here below (Fig. R1). On the next page, we have explained in more detail what we think might have occurred during the computational analysis for the Daigaku et al. manuscript that caused such a shift in their data.

Fig. R1

Data from library GSM1519714
 According to Keszthelyi et al., *Nat. Protocols* 2015

With Trimming	A	C	G	U/T
0	0.29448	0.33240	0.23042	0.14271
+1	0.28231	0.15008	0.22337	0.34424

In the data analysis section of Keszthelyi et al.'s *Nature Protocols* 2015 publication, the authors specify that the 5'-most end of the reads should be trimmed by one nucleotide (--trim5 1) prior to alignment (please see code below as shown in step 63 of their manuscript (doi:10.1038/nprot.2015.116)).

```
filepath/bowtie2 -x filepath/index_name --trim5 1 --trim3 30 -1 pol-d_R1.fastq -  
2 pol-d-R2.fastq -S pol-d.sam
```

In the field of rNMP mapping, the 5'-most position of the aligned read is *critical*, because this is from where the chromosomal coordinates of rNMPs are calculated. For example, for Pu-seq data, the rNMP is located one nucleotide upstream from the aligned 5'-most nucleotide of the read. Based on our analysis, we found that when the 5'-most nucleotides of the reads are trimmed according to the instructions given in Keszthelyi et al.'s *Nature Protocols* manuscript, the patterns at the +1 position in Supplementary Figure 3 of Daigaku et al.'s manuscript match those of the rNMP position (please refer to position 0 of Fig. R1 provided above). Since we obtain the same patterns as Daigaku et al. when we do NOT trim, but shifted results when we DO trim, it is evident that the positions of the rNMPs are shifted and the rNMP patterns for Pu-seq actually match those of ribose-seq and Zhou et al.'s *Nature Communications* 2019 study with low rU.

REVIEWERS' COMMENTS:

Reviewer #4 (Remarks to the Author):

Balachander et al. use the ribose-seq method to analyze the identity and distribution of rNMPs in nuclear and mitochondrial yeast genomes. The authors prepare and analyze 34 mitochondrial and 25 nuclear libraries from three distinct yeast species, including WT and *rnh1* and *rnh2* mutant derivatives from several different yeast strains. Based on the analysis the authors confirm that rNMPs are not distributed randomly throughout the genome, finding that the nucleotide immediately upstream of the rNMP has a strong effect on ribonucleotide positioning. The authors conclude that rC and rG are most frequently incorporated, and the upstream nucleotide is most frequently deoxyadenosine.

To verify that the obtained results are not an artefact of the ribose-seq technique (primary concern of reviewer 2) the authors analyzed five additional *S. cerevisiae* *rnh201* mitochondrial and nuclear libraries prepared using the emRiboSeq method. Consistent with the authors conclusions, results in Fig. 3A and Supp. Fig. 4C indicate striking similarity of dinucleotide frequencies in ribose-seq and emRiboSeq mitochondrial libraries. However, similarity between the dinucleotide frequency heatmaps for ribose-seq and EmRiboSeq nuclear libraries (Fig. 3b and Supp. Fig.4D) is less obvious. The heatmap for the emRiboSeq nuclear library suggests that dC, and not dA, is most frequently upstream from rC and rG. The frequency of dArC and dArG is not different from the actual % of AC and AG, respectively. Despite this there are common trends in the dinucleotide frequency heatmaps of emRiboSeq and riboseseq nuclear libraries. However, the authors should discuss in more detail why they consider these results to be in good agreement.

To understand the reason for the discrepancy in rU abundance in the *S. pombe* genome between the authors' results and published data (Diagaku et al. 2015), pointed out by reviewer 2 as a concern, the authors re-analyzed one of the published libraries and found an error in the computational analyses of the published data, showing that the published data are in fact consistent with their results.

Finally, I totally agree with the authors that there is a big gap between expecting a result and providing experimental support for "the expected" result and as the authors say and show here, bridging this gap involves extensive efforts.

Overall I do not believe that reviewer 2's concerns invalidate the authors main conclusions.

Page 6, line 21, Fig. 1C,D. There is no panel C or D in Figure 1.

Page 39, legend to Supplementary Table 4, line 24, "...S. pombe mitochondrial..." Should be nuclear.

Page 43, legend to Supplementary Figure 4, figure title is not accurate .

There is a discrepancy between description (D), (E) and the annotation in the figure.

March 25, 2020

Point-by-point Response to Reviewer #4, NCOMMS-19-18835C

1) To verify that the obtained results are not an artefact of the ribose-seq technique (primary concern of reviewer 2) the authors analyzed five additional *S. cerevisiae* rnh201 mitochondrial and nuclear libraries prepared using the emRiboSeq method. Consistent with the authors conclusions, results in Fig. 3A and Supp. Fig. 4C indicate striking similarity of dinucleotide frequencies in ribose-seq and emRiboSeq mitochondrial libraries. However, similarity between the dinucleotide frequency heatmaps for ribose-seq and EmRiboSeq nuclear libraries (Fig. 3b and Supp. Fig.4D) is less obvious. The heatmap for the emRiboSeq nuclear library suggests that dC, and not dA, is most frequently upstream from rC and rG. The frequency of dArC and dArG is not different from the actual % of AC and AG, respectively. Despite this there are common trends in the dinucleotide frequency heatmaps of emRiboSeq and riboseseq nuclear libraries. However, the authors should discuss in more detail why they consider these results to be in good agreement.

-We thank the Reviewer for this suggestion. As the Reviewer stated, the dinucleotide heatmaps in Supplementary Figure 4B,D,F,I,J show common trends in the dinucleotide frequencies between the emRiboSeq and ribose-seq nuclear libraries, providing evidence that our results are valid rather than an artifact of the technique. Overall, our ribose-seq libraries show increased frequencies of dArC and dArG relative to the corresponding background frequencies, with some strain backgrounds showing larger increases than others. Although the dArC and dArG frequencies are not as increased to the same degree in these particular emRiboSeq libraries as in some of our ribose-seq libraries, these emRiboSeq libraries show trends in relative dinucleotide frequencies that are similar to many of our ribose-seq libraries. It is important to note that these emRiboSeq libraries (from Reijns et al. *Nature* 2015) were prepared from only one particular strain background ($\Delta I(-2)I-7BYUNI300$ derived from $\Delta 7$) of *S. cerevisiae*. Since only one strain background of emRiboSeq data is publicly available, we could not evaluate overall trends across different yeast strains with emRiboSeq as we did with ribose-seq. If we could analyze additional emRiboSeq libraries from different yeast strain backgrounds, it is likely that we would observe similar variation as we do with ribose-seq libraries. To explain this point in more detail, we have updated the revised manuscript on page 9, lines 11-14: << Analysis of the five *rnh201*-null emRiboSeq libraries, which are all from the same strain background ($\Delta I(-2)I-7BYUNI300$), revealed common trends in the sequence context of rNMPs observed in the ribose-seq libraries of the same genotype. In particular, the strongest biases were seen for the -1 dNMP (Supplementary Fig. 4B,D,F,I,J).>>

2) To understand the reason for the discrepancy in rU abundance in the *S. pombe* genome between the authors' results and published data (Diagaku et al. 2015), pointed out by reviewer 2 as a concern, the authors re-analyzed one of the published libraries and found an error in the computational analyses of the published data, showing that the published data are in fact consistent with their results.

-We thank the Reviewer for acknowledging our work in uncovering the error in the previously published Pu-seq results and the consistency between our results and the corrected Pu-seq results.

3) Finally, I totally agree with the authors that there is a big gap between expecting a result and providing experimental support for “the expected” result and as the authors say and show here, bridging this gap involves extensive efforts.

Overall I do not believe that reviewer 2’s concerns invalidate the authors main conclusions.

-We thank the Reviewer for acknowledging our substantial efforts on this work and his/her support of the validity of our manuscript’s main conclusions.

4) Page 6, line 21, Fig. 1C,D. There is no panel C or D in Figure 1.

-We thank the Reviewer for bringing this to our attention. We have updated the manuscript accordingly on page 5, from line 23 of the revised manuscript: << Data analysis from the *rnh1*-null libraries showed rNMP content similar to that of mitochondrial wild-type and *rnh201*-null cells of E134 (Fig. 2, Supplementary Figure 2C,D and Supplementary Data 1)>>.

5) Page 39, legend to Supplementary Table 4, line 24, “....*S. pombe* mitochondrial....” Should be nuclear.

-We thank the Reviewer for bringing this to our attention. Supplementary Table 4 is Supplementary Data 2 in the revised version. We have updated this table legend accordingly: <<(C) According to the PomBase genome database, the A+T content in *S. pombe* mtDNA is 69.91%, with 34.95% A, 34.95% T, 15.045% C and 15.045% G; while the A+T content in *S. pombe* nDNA is 63.95%, with 31.975% A, 31.975% T, 18.025% C and 18.025% G.>>

6) Page 43, legend to Supplementary Figure 4, figure title is not accurate. There is a discrepancy between description (D), (E) and the annotation in the figure.

-We thank the Reviewer for bringing this to our attention. We have modified the title of Supplementary Figure 4 to: << Supplementary Figure 4. Impact of dNMP positions on rNMP occurrence in mtDNA and nDNA of emRiboSeq libraries >>, and part of its legend to: <<Heatmap analyses with normalized frequency of (C) NR, (E) RN, (G) N-R and (H) R-N for mitochondrial emRiboSeq libraries, and (D) NR, (F) RN, (I) N-R and (J) R-N for nuclear emRiboSeq libraries>>.